# BETTER TEACHER BETTER STUDENT: DYNAMIC PRIOR KNOWLEDGE FOR KNOWLEDGE DISTILLATION

**Martin Zong**[*1], **Zengyu Qiu**[*1], **Xinzhu Ma**[*2,3], **Kunlin Yang**[*†1], **Chunya Liu**[1],
**Jun Hou**[1], **Shuai Yi**[1], and **Wanli Ouyang**[2,3]
[1]SenseTime Research, [2]Shanghai AI Lab, [3]The University of Sydney
`{zongdaoming,qiuzengyu,yangkunlin,liuchunya,houjun,yishuai}@sensetime.com,`
`{xinzhu.ma, wanli.ouyang}@sydney.edu.au`

## ABSTRACT

Knowledge distillation (KD) has shown very promising capabilities in transferring learning representations from large models (teachers) to small models (students). However, as the capacity gap between students and teachers becomes larger, existing KD methods fail to achieve better results. Our work shows that the 'prior knowledge' is vital to KD, especially when applying large teachers. Particularly, we propose the dynamic prior knowledge (DPK), which integrates part of teacher's features as the prior knowledge before the feature distillation. This means that our method also takes the teacher's feature as 'input', not just 'target'. Besides, we dynamically adjust the ratio of the prior knowledge during the training phase according to the feature gap, thus guiding the student in an appropriate difficulty. To evaluate the proposed method, we conduct extensive experiments on two image classification benchmarks (*i.e.* CIFAR100 and ImageNet) and an object detection benchmark (*i.e.* MS COCO). The results demonstrate the superiority of our method in performance under varying settings. Besides, our DPK makes the performance of the student model positively correlated with that of the teacher model, which means that we can further boost the accuracy of students by applying larger teachers. More importantly, DPK provides a fast solution in teacher model selection for any given model. Our code will be released at `https://github.com/Cuibaby/DPK`.

## 1 INTRODUCTION

Tremendous efforts have been made in crafting lightweight deep neural networks applicable to the real-world scenarios. Representative methods include network pruning (He et al., 2017), model quantization (Habi et al., 2020), neural architecture search (NAS) (Wan et al., 2020), and knowledge distillation (KD) (Bucilua et al., 2006; Hinton et al., 2015), *etc*. Among them, KD has recently emerged as one of the most flourishing topics due to its effectiveness (Liu et al., 2021a; Zhao et al., 2022; Chen et al., 2021; Heo et al., 2019a) and wide applications (Yang et al., 2022a; Chong et al., 2022; Liu et al., 2019a; Yim et al., 2017b; Zhang & Ma, 2020).

Particularly, the core idea of KD is to transfer the distilled knowledge from a well-performed but cumbersome teacher to a compact and lightweight student. Based on this, numerous methods have been proposed and achieved great success. However, with the deepening of research, some related issues are also discussed. In particular, several works (Cho & Hariharan, 2019; Mirzadeh et al., 2020; Hinton et al., 2015; Liu et al., 2021a) report that with the increase of teacher model in performance, the accuracy of student gets saturated (which might be unsurprising). To make matters worse, when playing the role of a teacher, the large teacher models lead to *significantly worse* performance than the relatively smaller ones. For example, as shown in Fig.1, ICKD (Liu et al., 2021a), a strong baseline model which also points out this issue, performing better under the guidance of small teacher models, whereas applying a large model as the teacher would considerably degrade the student performance.

Same as (Cho & Hariharan, 2019; Mirzadeh et al., 2020), we attribute the cause of this issue to the capacity gap between the teachers and the students. More specifically, the small student is

---
[*]Equal contribution. [†]Corresponding author.

hard to 'understand' the high-order semantics extracted by the large model. This problem will be exacerbated when applying larger teachers, and it makes the student's accuracy inversely correlated with the capacity of the teacher model[1]. Note that this problem also exists for humans, and human teachers often tell students some *prior knowledge* to facilitates their learning in this case. Moreover, the experienced teachers can also *adjust* the amounts of provided prior knowledge accordingly for different students to fully develop their potentials.

Inspired by the above observations from human teachers, we propose the dynamic prior knowledge (DPK) framework for feature distillation. Specifically, to provide the prior knowledge to the student, we replace student's features at some random spatial positions with corresponding teacher features at the same positions. Besides, we further design a ViT (Dosovitskiy et al., 2020)-style module to fully integrate this 'prior knowledge' with student's features. Furthermore, our method also dynamically adjusts the amounts of the prior knowledge, reflected in the proportion of teacher features in the hybrid feature maps. Particularly, DPK dynamically computes the differences of features between the student and the teacher in the training phase, and updates the ratio of feature mixtures accordingly. In this way, students always learn from the teacher at an appropriate difficulty, thus alleviating the performance degradation issue.

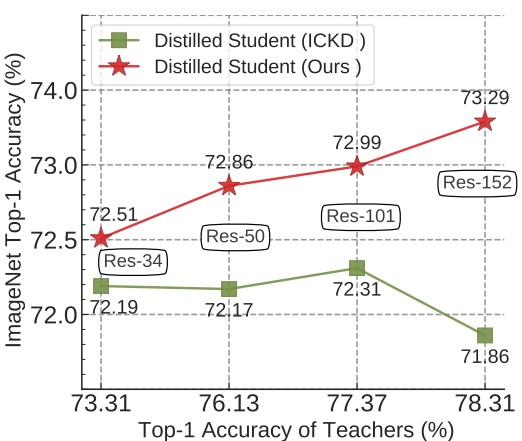

Figure 1: **Top-1 accuracy of ResNet-18 w.r.t. various teachers on ImageNet.** Different from the baseline model (ICKD (Liu et al., 2021a)), our method shows better performance and makes the performance of student positively correlated with that of the teacher.

We evaluate DPK on two image classification benchmarks (*i.e.* CIFAR-100 (Krizhevsky et al., 2009) and ImageNet (Deng et al., 2009)) and an object detection benchmark (*i.e.* MS COCO (Lin et al., 2014)). Experimental results indicate that DPK outperforms other baseline models under several settings. More importantly, our method can be further improved by applying larger teachers (see Fig. 1 for an example). We argue that this characteristic of DPK not only further boosts student performance, but also provides a quick solution in model selection for finding the best teacher for a given student. In addition, we conduct extensive ablations to validate each design of DPK in detail.

In summary, the main contributions of this work are:

- We propose the prior knowledge mechanism for feature distillation, which can fully excavate the distillation potential of big models. To the best of our knowledge, our method is the first to take the features of teachers as 'input', not just 'target' in knowledge distillation.
- Based on our first contribution, we further propose the dynamic prior knowledge (DPK). Our DPK provides a solution to the *'larger models are not always better teachers'* issue. Besides, it also gives better (or comparable) results under several settings.

## 2 METHODOLOGY

In this section, we first provide the background of KD, and then introduce the framework and details of the proposed DPK.

### 2.1 PRELIMINARY

The existing KD methods can be grouped into two groups. In particular, the logits-based KD methods distill the *dark knowledge* from the teacher by aligning the soft targets between the student and

---

[1]student's performance is positively correlated with the accuracy of the teacher if the capacity gap is fixed, see Appendix A.6 for more details.

teacher, which can be formulated as a loss item:

$$\mathcal{L}_{logits} = \mathcal{D}_{logits}(\sigma(\boldsymbol{z^s}; \tau), \sigma(\boldsymbol{z^t}; \tau)), \tag{1}$$

where $\boldsymbol{z}^s$ and $\boldsymbol{z}^t$ are the logits from the student and the teacher. $\sigma(\cdot)$ is the softmax function that produces the category probabilities from the logits, and $\tau$ is a non-negative temperature hyper-parameter to scale the smoothness of predictive distribution. Specifically, we have $\sigma_i(\boldsymbol{z}; \tau) = \text{softmax}(\exp(z_i/\tau))$. $\mathcal{D}_{logits}$ is a loss function which can capture the difference between two categorial distributions, *e.g.* Kullback-Leibler divergence. Similarly, the feature-based KD methods, whose main idea is to mimick the feature representations between students and teachers, can also be represented as an auxiliary loss item:

$$\mathcal{L}_{feat} = \mathcal{D}_{feat}(\mathsf{T}_s(\mathbf{F}^s), \mathsf{T}_t(\mathbf{F}^t)), \tag{2}$$

where $\mathbf{F}^s$ and $\mathbf{F}^t$ denote the feature maps from the student and the teacher, respectively. $\mathsf{T}_s, \mathsf{T}_t$ denote the student and the teacher transformation module respectively, which align the dimensions of $\mathbf{F}^s$ and $\mathbf{F}^t$ (and transform the feature representations, such as (Tung & Mori, 2019)). $\mathcal{D}_{feat}$ denotes the function which can compute the distance between two feature maps, such as $\ell_1$- or $\ell_2$-norm. So the KD methods can be represented by a generic paradigm. The final loss is the weighted sum of the classification loss $\mathcal{L}_{cls}$ (the original training loss), the logits distillation loss, and the feature distillation loss:

$$\mathcal{L} = \mathcal{L}_{cls} + \alpha\mathcal{L}_{logits} + \beta\mathcal{L}_{feat}, \tag{3}$$

where $\{\alpha, \beta\}$ are hyper-parameters controlling the trade-off between these three losses.

## 2.2 DYNAMIC PRIOR KNOWLEDGE

An overview of the proposed DPK is presented in Fig. 2. As a feature distillation method, the main contributions of the DPK include the prior knowledge mechanism and dynamic mask generation. Here we give the details of these two designs.

**Prior knowledge.** To introduce the prior knowledge, the teacher provides part of its features to the student, and the Eq. 2 can be reformulated as:

$$\mathcal{L}_{feat} = \mathcal{D}_{feats}(\mathsf{T}_s(\mathbf{F}^s, \mathbf{F}^t), \mathsf{T}_t(\mathbf{F}^t)), \tag{4}$$

and then we introduce how to build the hybrid feature map, *i.e.* the student transformation module $\mathsf{T}_s$. Specifically, given the paired feature maps $\mathbf{F}^s$ and $\mathbf{F}^t$ from the student and the teacher, we divide them into several non-overlapping patches using a $k \times k$ convolution with stride $k$, where $k$ is the pre-defined size of the patches. Meanwhile, we also align the dimension of the features with this convolution layer. Then we randomly mask a subset of feature patches of the student under an uniform distribution with the ratio $\pi$, and also generate the complementary feature patches from the teachers. We take these feature patches as *token sequences* and use two ViT (Dosovitskiy et al., 2020)-based encoders to further process them. After that, we stitch these token sequences together to generate the hybrid features (tokens), add the positional embedding to these hybrid tokens, and further integrate them with another ViT-based decoder. Finally, we re-organize generated token sequences into original shape and apply the feature distillation loss on the hybrid features and original teacher features. The transformation module $\mathsf{T}_t$ for teacher feature is an identical mapping.

**Dynamic mechanism.** In the above solution, we mix the features of teacher and student with the hyper-parameter $\pi$. Furthermore, we empirically find that: (*i*) the optimal $\pi$ for different model combinations is different (see Table 5 for related experiments), and (*ii*) the feature gap is different in the early and late stages of training phase. These facts inspire us to adjust the value of the masking ratio flexibly and dynamically according to the teacher-student gap, *e.g.* when the teacher-student gap is large, students need more prior knowledge to guide them. Upholding this principle, given a minibatch of feature sets for a teacher-student pair, we flatten them to yield $\mathbf{F}_t \in \mathbb{R}^{B \times \mathsf{c} \times \mathsf{h} \times \mathsf{w}} \mapsto \mathbf{X} \in \mathbb{R}^{B \times p_1}$ and $\mathbf{F}_s \in \mathbb{R}^{B \times \mathsf{c} \times \mathsf{h} \times \mathsf{w}} \mapsto \mathbf{Y} \in \mathbb{R}^{B \times p_2}$. We then set the iteration-specific masking ratio $\pi$ at the $i^{\text{th}}$ minibatch as :

$$\pi_i = 1 - \text{CKA}_{\text{minibatch}}(\mathbf{X}_i, \mathbf{Y}_i), \tag{5}$$

where $i$ indexes the minibatch within a training epoch. $\text{CKA}_{\text{minibatch}}$ Nguyen et al. (2020) is the minibatch version of the Centered Kernel Alignment (CKA) Kornblith et al. (2019), which is representation similarity measure that is widely used for quantitative understanding the representations

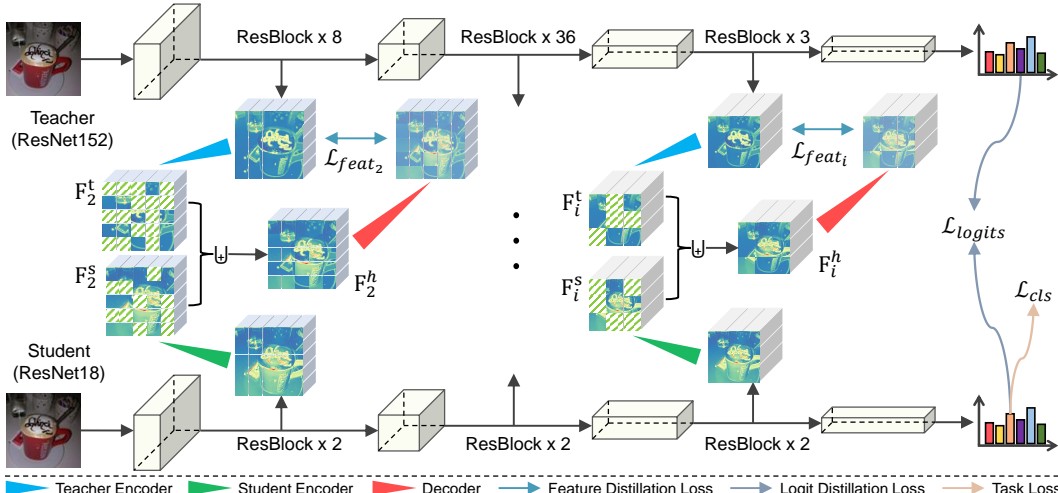

Figure 2: **Illustration of the proposed DPK.** For each feature distillation stage, the student feature map and the teacher feature map are sent to corresponding encoders to generate the $\mathbf{F}^s$ and $\mathbf{F}^t$. Then, a subset of student feature patches is replaced by that of the teacher (the ⊎ denotes the feature stitching operation). After that, DPK further integrates the hybrid feature $\mathbf{F}^h$ with a decoder before applying feature distillation loss. Note that the proportion of $\mathbf{F}^s$ and $\mathbf{F}^t$ in $\mathbf{F}^h$ is dynamically generated, which is omitted in this figure.

learned by neural networks. Specifically, CKA takes the two feature representations $\mathbf{X}$ and $\mathbf{Y}$ as input and computes their normalized similarity in terms of the Hilbert-Schmidt Independence Criterion (HSIC). $\text{CKA}_{\text{minibatch}}$ instead uses an unbiased estimator of HSIC Song et al. (2012), called $\text{HISC}_1$,

$$\text{HSIC}_1(\mathbf{K}, \mathbf{L}) = \frac{1}{n(n-3)}\left(\text{tr}(\tilde{\mathbf{K}}\tilde{\mathbf{L}}) + \frac{\mathbf{1}^\top\tilde{\mathbf{K}}\mathbf{1}\mathbf{1}^\top\tilde{\mathbf{L}}\mathbf{1}}{(n-1)(n-2)} - \frac{2}{n-2}\mathbf{1}^\top\tilde{\mathbf{K}}\tilde{\mathbf{L}}\mathbf{1}\right),\tag{6}$$

where $\tilde{\mathbf{K}}$ and $\tilde{\mathbf{L}}$ are obtained by setting the diagonal entries of similarity matrices $\mathbf{K}$ and $\mathbf{L}$ to zero. Then $\text{CKA}_{\text{minibatch}}$ can be computed by averaging $\text{HISC}_1$ scores over $k$ minibatches:

$$\text{CKA}_{\text{minibatch}} = \frac{\frac{1}{k}\sum_{i=1}^k \text{HSIC}_1(\mathbf{X}_i\mathbf{X}_i^\top, \mathbf{Y}_i\mathbf{Y}_i^\top)}{\sqrt{\frac{1}{k}\sum_{i=1}^k \text{HSIC}_1(\mathbf{X}_i\mathbf{X}_i^\top, \mathbf{X}_i\mathbf{X}_i^\top)}\sqrt{\frac{1}{k}\sum_{i=1}^k \text{HSIC}_1(\mathbf{Y}_i\mathbf{Y}_i^\top, \mathbf{Y}_i\mathbf{Y}_i^\top)}},\tag{7}$$

where $\mathbf{X}_i \in \mathbb{R}^{B\times p_1}$ and $\mathbf{Y}_i \in \mathbb{R}^{B\times p_2}$ are now matrices containing activations of the $i^{\text{th}}$ minibatch of $B$ examples sampled without replacement.

**Remarks.** Eq. (7) allows us to efficiently and robustly estimate feature dissimilarity between the teacher-student pair using minibatches. A lower CKA indicates a greater feature gap between students and teachers. And the higher the masking ratio, the larger the feature regions masked by the student and the larger the countparts provided by the teacher. Eq. (5) naturally establishes the connection between feature gap and masking ratio, with its effectiveness and design choices verfied in Sec. 3.3.

Due to the space limitation, we only introduce the main designs of our DPK, and more details (*e.g.* how to apply DPK in object detection) can be found in Appendix A.2.

## 3 EXPERIMENTS

We conduct extensive experiments on *image classification* and *object detection*. Moreover, we present various *ablations* and *analysis* for the proposed method. Besides, our codes as well as training recipes will be publicly available for reproducibility.

### 3.1 IMAGE CLASSIFICATION

We evaluate our method on CIFAR-100 and ImageNet for image classification (see Appendix A.1 for the introduction of these datasets and related evaluation metrics). We compare DPK with a wide

range of baseline models, including KD (Hinton et al., 2015), FitNets (Romero et al., 2015), FT (Kim et al., 2018), AB (Heo et al., 2019b), AT (Zagoruyko & Komodakis, 2017), PKT (Passalis & Tefas, 2018), SP (Tung & Mori, 2019), SAD (Ji et al., 2021), CC (Peng et al., 2019), RKD (Park et al., 2019), VID (Ahn et al., 2019), CRD (Tian et al., 2020), OFD (Heo et al., 2019a), ReviewKD (Chen et al., 2021), DKD (Zhao et al., 2022), ICKD-C (Liu et al., 2021a) and MGD (Yang et al., 2022b).

**Results on CIFAR-100.** We first evaluate DPK on the CIFAR-100 dataset and summarize the results on Table 1. From these results, we can observe that the proposed DPK performs best for all six teacher-student pairs, which firmly demonstrates the effectiveness of our method.

Table 1: **Results on the CIFAR-100 validation set.** We report the top-1 accuracy (%) of the methods for *homogeneous* teacher-student pairs. "-" indicates results are not available, and we highlight the best results in **bold**.

| Teacher | WRN40-2 | WRN40-2 | ResNet56 | ResNet110 | ResNet110 | VGG13 |
|---|---|---|---|---|---|---|
| Acc. | 75.61 | 75.61 | 72.34 | 74.31 | 74.31 | 74.64 |
| Student | WRN16-2 | WRN40-1 | ResNet20 | ResNet20 | ResNet32 | VGG8 |
| Acc. | 73.26 | 71.98 | 69.06 | 69.06 | 71.14 | 70.36 |
| KD | 74.92 | 73.54 | 70.66 | 70.67 | 73.08 | 72.98 |
| FitNets | 73.58 | 72.24 | 69.21 | 68.99 | 71.06 | 71.02 |
| AT | 74.08 | 72.77 | 70.55 | 70.22 | 72.31 | 71.43 |
| PKT | 74.54 | 73.45 | 70.34 | 70.25 | 72.61 | 72.88 |
| SP | 73.83 | 72.43 | 69.67 | 70.04 | 72.69 | 72.68 |
| CC | 73.56 | 72.21 | 69.63 | 69.48 | 71.48 | 70.71 |
| RKD | 73.35 | 72.22 | 69.61 | 69.25 | 71.82 | 71.48 |
| VID | 74.11 | 73.30 | 70.38 | 70.16 | 72.61 | 71.23 |
| CRD | 75.48 | 74.14 | 71.16 | 71.46 | 73.48 | 73.94 |
| OFD | 75.24 | 74.33 | 70.98 | - | 73.23 | 73.95 |
| ReviewKD | 76.12 | 75.09 | 71.89 | - | 73.89 | 74.84 |
| DKD | 76.24 | 74.81 | 71.97 | - | 74.11 | 74.68 |
| ICKD-C | 75.57 | 74.63 | 71.69 | 71.91 | 74.11 | 73.88 |
| **DPK** | **76.42** | **75.27** | **72.37** | **72.44** | **74.89** | **74.96** |

**Results on ImageNet.** We also conduct experiments on the large-scale ImageNet to evaluate our DPK. In particular, following the previous conventions (Tian et al., 2020; Chen et al., 2021), we present the performance of ResNet-18 guided by ResNet-34 in Table 2. The results show the superiority of DPK in performance to other baselines.

**KD for heterogeneous models.** Table 1 and 2 show the experiments for homogeneous models (*e.g.* ResNet18 and ResNet34), and we show our method can also be applied to heterogeneous models (*e.g.* MobileNet and ResNet50) in this part. From the results shown in Table 3, we can observe that DPK performs best among all listed methods for heterogeneous models (more experiments for this setting can be found in Appendix A.4).

**Better teacher, better student.** The above experiments show DPK performs well for the common teacher-student pairs. Here we show our method can be further improved with better teachers. As illustrated in Fig. 3, the accuracy of the student model trained by our method is continuously improved by progressively replacing larger teacher models, while the model trained by other algorithms fluctuates in performance. Meanwhile, our method surpasses it counterparts at each stage and progressively widen the performance gap. To show the generalization, we also present the performance change for other teacher-student combinations in Fig. 4, which confirms the same conclusion again. Note that it is a reasonable phenomenon that the performance of the given student tends to be saturated gradually, but the performance fluctuation will cause many difficulties in practical application.

## 3.2 OBJECT DETECTION

DPK can also be applied to other tasks, and we evaluate it on a popular one, *i.e.* object detection. Note that our method can be easily integrated into other KD methods, and we apply DPK into FGD (Yang et al., 2022a) and evaluate the performance on the most commonly used MS COCO dataset (Lin et al., 2014) (see Appendix A.1 and A.2 for the details of dataset/metrics and implementation).

**Comparison with SOTA methods.** As presented in Table 4, we evaluate our model on a one-stage detector (RetinaNet (Lin et al., 2017b)) and a two-stage detector (Faster-RCNN (Ren et al., 2015)) with several strong baselines, including FGFI (Wang et al., 2019), GID (Dai et al., 2021), FGD

Table 2: **Results on ImageNet validation set.** We show the top-1 and top-5 accuracy (%) for ResNet18 guided by ResNet34. "-" indicates the results are not available.

| Acc. | Student | Teacher | KD | RKD | CRD+KD | SAD | CC | ICKD-C | ReviewKD | DKD | MGD | Ours |
|---|---|---|---|---|---|---|---|---|---|---|---|---|
| Top-1 | 69.55 | 73.31 | 70.68 | 71.34 | 71.38 | 71.38 | 70.74 | 72.19 | 71.61 | 71.70 | 71.80 | **72.51** |
| Top-5 | 89.09 | 91.42 | 90.16 | 90.37 | - | 90.49 | - | 90.72 | 90.51 | 90.41 | 90.40 | **90.77** |

Table 3: **Results for heterogeneous models.** We show the top-1 and top-5 accuracy (%) of MobileNetV2 guided by ResNet50 on ImageNet validation set.

| Acc. | Student | Teacher | FT | AB | AT | OFD | CRD | ReviewKD | KD | DKD | MGD | Ours |
|---|---|---|---|---|---|---|---|---|---|---|---|---|
| Top-1 | 68.87 | 76.16 | 69.88 | 69.89 | 69.56 | 71.25 | 71.37 | 72.56 | 68.58 | 72.05 | 72.59 | **73.26** |
| Top-5 | 88.76 | 92.86 | 89.50 | 88.71 | 89.33 | 90.34 | 90.41 | 91.00 | 88.98 | 91.05 | 90.94 | **91.17** |

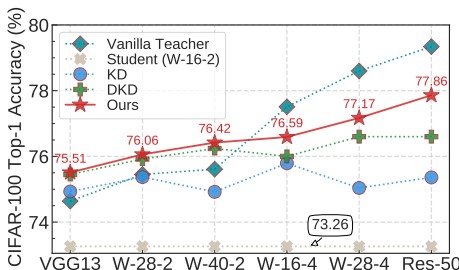 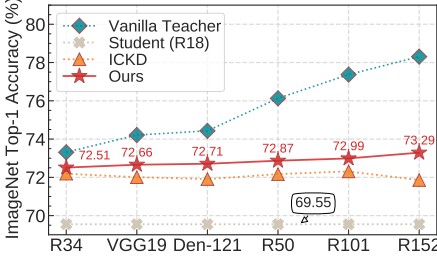

Figure 3: **Better teacher, better student.** We show the top-1 accuracy of our method and some baselines on the CIFAR-100 (*left*) and ImageNet (*right*).

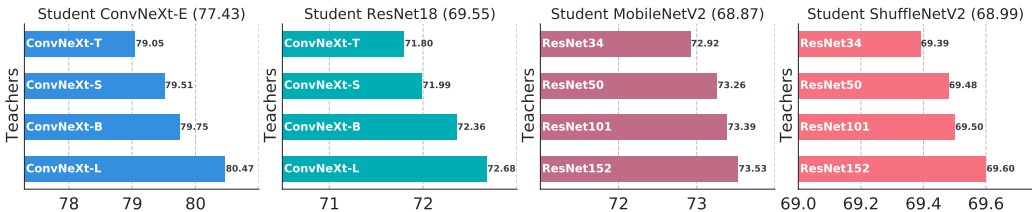

Figure 4: **More teacher-student combinations**, covering the lightweight ShuffleNetV2 (Ma et al., 2018) to the recently ConvNeXt (Liu et al., 2022). Top-1 accuracy (%) on ImageNet val set is reported. Both *homogeneous* and *heterogeneous* settings are considered. By changing the teacher size, we see that, *larger models are always better teachers* holds true when applying DPK. ConvNeXt-E is a small model built by us, following the design principles of ConvNeXt (Liu et al., 2022), see Appendix A.2 for details. Best viewed in color with zoom in.

(Yang et al., 2022a), and DeFeat (Guo et al., 2021). Following the previous conventions, we first adopt the ResNet101-FPN as the teacher model and ResNet50-FPN as the student model, and our model achieves better (or matched) performance than other baselines, *i.e.* our model gets a matched performance with FGD, and significantly surpasses other baselines in accuracy.

**Better teacher, better student.** Similar to image classification, our method also benefits from better teacher model for object detection. In particular, we enlarge the capacity gap between the teacher and student models, and the results in Table 4 suggest that effectiveness of our method can be further improved by replacing more powerful teacher models. For instance, replacing teacher model from ResNet101-FPN to ResNet152-FPN further improves the student's performance (ResNet50-FPN, Faster RCNN) by 0.6 mAP, while the number for FGD (Yang et al., 2022a) is 0.1 for the same setting. This conclusion also stands for other teacher-student pairs and frameworks.

## 3.3 ABLATION STUDIES

In this section, we provide extensive ablation studies to analyze the effects of each component of DPK. The experiments are conducted on ImageNet for classification task, ResNet34 and ResNet18 are adopted as teacher and student. Only last stage distillation is applied unless stated otherwise.

Table 4: **Results on object detection**. Experiments are evaluated on COCO validation set. 'T' and 'S' represent the teacher and the student, respectively. "-" indicates results are not available.

| Methods | RetinaNet | | | | | | Faster-RCNN | | | | | |
|---|---|---|---|---|---|---|---|---|---|---|---|---|
| | mAP | AP$_{50}$ | AP$_{75}$ | AP$_S$ | AP$_M$ | AP$_L$ | mAP | AP$_{50}$ | AP$_{75}$ | AP$_S$ | AP$_M$ | AP$_L$ |
| R101-FPN(T) | 38.9 | 58.0 | 41.5 | 21.0 | 42.8 | 52.4 | 39.9 | 60.1 | 43.3 | 23.5 | 44.2 | 51.5 |
| R50-FPN(S) | 37.4 | 56.7 | 39.6 | 20.6 | 40.7 | 49.7 | 38.4 | 59.0 | 42.0 | 21.5 | 42.1 | 50.3 |
| FGFI | 38.6 | 58.7 | 41.3 | 21.4 | 42.5 | 51.5 | 39.3 | 59.8 | 42.9 | 22.5 | 42.3 | 52.2 |
| GID | 39.1 | 59.0 | 42.3 | 22.8 | 43.1 | 52.3 | 40.2 | 60.7 | 43.8 | 22.7 | 44.0 | 53.2 |
| FGD | 39.6 | - | - | 22.9 | 43.7 | 53.6 | 40.4 | - | - | 22.8 | 44.5 | 53.5 |
| **Ours** | **39.7** | 58.6 | 42.5 | 22.8 | 43.6 | 53.6 | **40.6** | 60.7 | 44.4 | 22.8 | 44.7 | 54.0 |
| R152-FPN(T) | 39.9 | 59.4 | 42.7 | 23.5 | 44.2 | 51.5 | 41.6 | 62.3 | 45.4 | 23.3 | 46.1 | 53.6 |
| R50-FPN(S) | 37.4 | 56.7 | 39.6 | 20.6 | 40.7 | 49.7 | 38.4 | 59.0 | 42.0 | 21.5 | 42.1 | 50.3 |
| FGFI | 38.9 | - | - | 21.9 | 42.5 | 52.2 | 39.9 | - | - | 22.9 | 43.6 | 52.8 |
| DeFeat | 39.7 | - | - | 23.4 | 43.6 | 52.9 | 40.9 | - | - | 23.6 | 44.8 | 53.5 |
| FGD | 39.7 | 58.9 | 42.9 | 23.2 | 44.0 | 52.7 | 40.5 | 61.1 | 44.2 | 23.4 | 44.4 | 53.7 |
| **Ours** | **40.2** | 59.6 | 43.0 | 23.4 | 44.5 | 53.6 | **41.2** | 61.6 | 45.0 | 23.3 | 45.2 | 54.3 |
| R101-FPN(T) | 38.9 | 58.0 | 41.5 | 21.0 | 42.8 | 52.4 | 39.9 | 60.1 | 43.3 | 23.5 | 44.2 | 51.5 |
| R18-FPN(S) | 33.1 | 51.4 | 34.9 | 17.4 | 35.8 | 43.4 | 33.8 | 53.2 | 36.8 | 19.2 | 36.6 | 43.6 |
| FGD | 34.7 | 52.5 | 37.1 | 17.8 | 38.0 | 48.5 | 36.0 | 55.8 | 39.2 | 18.3 | 39.4 | 48.7 |
| Ours | **35.8** | 53.9 | 38.2 | 19.1 | 39.4 | 49.1 | **36.7** | 56.2 | 40.1 | 19.2 | 40.3 | 49.2 |

Table 5: **Ablations on mask ratios.** We report top-1 accuracy on ImageNet with setting (*a*): ResNet18 as student, ResNet34 as teacher, and setting (*b*): ResNet18 as student, ResNet101 as teacher.

| setting | teacher | student | 15% | 35% | 55% | 75% | 95% | dynamic |
|---|---|---|---|---|---|---|---|---|
| (a) | 73.31 | 69.55 | 72.33 | 72.34 | **72.38** | 72.36 | 72.34 | **72.46** |
| (b) | 77.37 | 69.55 | 72.46 | 72.55 | 72.57 | **72.60** | 72.53 | **72.87** |

**Mask ratio.** Feature masking (and stitching) is a key component of our method, and Table 5 reports the results of various DPK variants under different mask ratios. Surprisingly, a *broad* range of masking ratios from 15% to 95% can offer considerable performance gains for students. This implies that the prior knowledge provided by teachers is very beneficial to students' network learning. Besides, note that the optimal mask ratio is *inconsistent* under different teacher-student pairs, *e.g.* 55% mask ratio performs best for ResNet-18 and ResNet34, while the optimal one for ResNet-18 and ResNe101 is 75%. Table 5 also shows students achieve the best accuracy using the proposed dynamic masking strategy, which suggests the necessity and effectiveness of the proposed *dynamic masking strategy* for automatic selection of mask ratio.

**Mask strategy.** Table 6 summarizes the effects of different mask strategies. ResNet34 and ResNet18 are adopted as teacher and student for all experiments in this part. For the fixed mask ratio, we take the simple random mask as baseline, and consider the `block-wise` mask strategy, introduced by BEIT (Bao et al., 2022). We also consider `grid-wise` mask, which regularly retains one of every four patches, similar to MAE (He et al., 2021). Besides fixed mask designs, several alternatives for realizing a dynamic mask ratio are explored. Particularly, the `cosinesimi` indicates that we use the cosine similarity to measure the teacher-student feature gap. The `exponential` decay schedule divides the mask ratio by the same factor every epoch, which can be expressed as $\pi_i = \pi_0 * (0.95)^{\text{epoch}_i}$, where $\pi_0$ is the initial mask ratio

| Masking Strategy | Ratio | Top-1 | Top-5 |
|---|---|---|---|
| Vanilla | ✗ | 69.55 | 89.09 |
| Random | 75% | 72.36 | 90.61 |
| Block-wise | 75% | 72.36 | 90.57 |
| Grid-wise | 75% | 72.32 | 90.38 |
| 1-CKA | ✗ | **72.46** | **90.67** |
| 1-CosineSimi | ✗ | 72.38 | 90.58 |
| Exponential decay | ✗ | 72.35 | 90.51 |
| Linear decay | ✗ | 72.36 | 90.58 |

Table 6: **Ablations on mask strategies.**

and is set to 1.0. The `linear` schedule decreases the mask ratio by the same decrement every epoch, which is defined as $\pi_i = \pi_0 - (\text{epoch}_i * \text{decrement})$, and we set $\text{decrement} = 0.95$. The results reveal that: *i*) simple random sampling works best for our DPK when using fixed mask ratios. *ii*) $1-\text{CKA}$ outperforms its competitors, and we use it to generate dynamic mask ratios by default. We also observe that both $1-\text{cosinesimi}$ and $1-\text{CKA}$ outperform the manually-set heuristic masking strategies, *e.g.*, `linear decay`, demonstrating the advantage of dynamically adjusting the ratio of teacher prior knowledge based on the teacher-student feature gap. Furthermore, the CKA is superior to the `cosinesimi`, indicating that our CKA can efficiently measure the similarity of

Table 7: **Ablations on prior knowledge.**

| Prior Knowledge | Top-1 | Top-5 |
|---|---|---|
| Zero-Padding | 72.16 | 90.35 |
| Learnable | 72.28 | 90.63 |
| Ours | **72.46** | **90.67** |

Table 8: **Ablations on transformation functions.**

| Transform | Top-1 | Top-5 |
|---|---|---|
| Baseline | 71.08 | 90.00 |
| Conv | 71.39 | 90.46 |
| MLP-Decoder | 72.31 | 90.50 |
| Encoder-Decoder | **72.46** | **90.67** |

Table 9: **Ablations on loss calculation.**

| Target | Top-1 | Top-5 |
|---|---|---|
| Non-Masked | 72.39 | 90.63 |
| Full | **72.46** | **90.67** |

the hidden representations between teachers and students using minibatches, and provide a robust way to automatically determine the masking ratio.

**Prior knowledge.** Table 7 ablates the significance of integrating teacher's knowledge in building hybrid student features (Eq. 4). Specifically, we use `zero-padding` and `learnable` mask token to play the role of teacher's feature in the masked position. The results show that no teacher-provided prior knowledge leads to worse performance, as it aggravates the burden of feature mimicking from students to teachers. This strongly confirms the effectiveness of offering students the prior knowledge from teachers via feature masking and stitching.

**Transformation module.** For feature-based distillation methods, the feature transformation modules $T_s$ and $T_t$ are required to convert the features into an easy-to-transfer form. We take FitNets (Romero et al., 2015) as baseline, which does not reduce the dimension of teacher's feature map and use a $1 \times 1$ convolutional layer to transform the feature dimension of the student to that of the teacher. In our method, we adopt the ViT-based encoder-decoder as the default transformation module. For further investigation, we also remove the encoder, and use a MLP layer to align the feature dimension (MLP-Decoder). Besides, the convolution-based encoder-decoder structure is also explored (Conv). More implementation details for these modules are deferred in the Appendix A.2. Table 8 shows the results, and we can observe that `encoder-decoder` reaches the best performance.

**Loss.** In the training phase, we apply the mean squared error (MSE) loss between the hybrid student features and teacher features in Eq. 4, while the loss can be on only the non-masked regions of student features, namely `non-masked`, or full feature maps, namely `full`. Table 9 shows the ablation results for these two settings. As can be seen, computing the loss on the full features performs better.

## 3.4 VISUALIZATIONS

In this part, we present some visualizations to show that our DPK does bridge the teacher-student gap in the feature-level. In particular, we visualize the feature similarity between ResNet18 and ResNet34 in Fig. 5. We can find that our DPK significantly improves the feature similarity (measured by CKA) between the student and the teacher. ICKD gets a lower similarity than the baseline, probably due to the fact that it models the feature relationships, instead of the features themselves. More related visualizations can be found in Appendix A.8, including other teacher-student combinations, the CKA curve in the training phase, and similarity maps measured by other metrics.

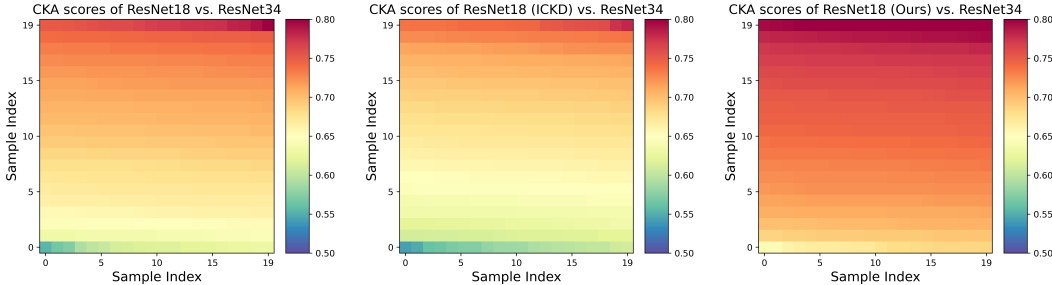

Figure 5: **CKA similarity between ResNet18 and ResNet34.** We visualize the CKA similarity between the original models (*left*), the models trained by ICKD (*middle*), and the models trained by our DPK (*right*). The experiments are conducted on the sampled ImageNet validation set (12,800 samples). We compute CKA with a batch size of 32 for the last stage, so these are 400 CKA values for each experiment. For better presentation, we rank these values and organize them as the heatmap representation. The larger the value, the more similar the features are.

## 4    RELATED WORK

**Knowledge distillation.** Existing studies on knowledge distillation (KD) can be roughly categorized into two groups: *logits-based* distillation and *feature-based* distillation. **The former**, pioneered by Hinton et al. (2015), is known as the classic KD, aiming to learn a compact student by mimicking the softmax outputs (logits) of an over-parameterized teacher. This line of work focuses on proposing effective regularization and optimization techniques (Zhang et al., 2018). Recently, DKD (Zhao et al., 2022) proposes to decouple the classical KD loss into two parts, *i.e.*, target class KD and non-target KD. Besides, several works (Phuong & Lampert, 2019; Cheng et al., 2020) also attempt to interpret the classical KD. **The latter**, represented by FitNet (Romero et al., 2015), encourages students to mimic the intermediate-level features from the hidden layers of teacher models. Since features are more informative than logits, feature-based distillation methods usually perform better than logits-based ones in the tasks that involve the localization information, such as object detection (Li et al., 2017; Wang et al., 2019; Dai et al., 2021; Guo et al., 2021; Yang et al., 2022a). This line of work mainly investigates what kinds of intermediate representations of features should be. These representations include singular value decomposition (Lee et al., 2018), attention maps (Zagoruyko & Komodakis, 2017), Gramian matrices (Yim et al., 2017a), gradients (Srinivas & Fleuret, 2018), pre-activations (Heo et al., 2019b), similarities and dissimilarities (Tung & Mori, 2019), instance relationships (Liu et al., 2019b; Park et al., 2019), inter-channel correlations (Liu et al., 2021a). A noteworthy work similar to DPK is AGNL (Zhang & Ma, 2020). In particular, AGNL has two attractive properties: *i*) attention-guided distillation, letting students' learning focus on the foreground objects and suppresses students' learning on the background pixels; and *ii*) non-local distillation, transferring the relation between different pixels from teachers to students. At a high level, both DPK and AGNL apply the masking strategy (random mask and attention-guided mask) and non-local relation modeling module (transformer and a self-designed non-local module).The key difference is that DPK integrates features of students and teachers with a dynamic mechanism.

**Performance degradation.** Prior works (Cho & Hariharan, 2019; Mirzadeh et al., 2020; Hinton et al., 2015; Liu et al., 2021a) also report that the performance of distilled student *degrades* when the gap between students and teachers becomes *large*. To solve this issue, ESKD (Cho & Hariharan, 2019) stops the teacher training early to make it under convergence and yield more softened logits. TAKD (Mirzadeh et al., 2020) introduces an extra intermediate-sized network termed *teacher assistant* to bridge the gap between teachers and students (more discussion for this method and our method can be found in Appendix A.3). Different from the above logits-based methods, our method directly reduces the gap between teachers and students in the feature space, and does not need extra intermediate models.

**Masked image modeling.** Emerged with the masked language modeling in NLP community, such as BERT (Devlin et al., 2018) and GPT (Radford et al., 2019; Brown et al., 2020), masked image modeling (MIM) (Pathak et al., 2016; Henaff, 2020) has gained increasing attention and shows promising potentials in representation learning. Particularly, MIM-based approaches generally *i*) divide an image or vedio into several non-overlapping patches or discrete visual tokens, *ii*) mask random subsets of these patches/tokens, and *iii*) predict the patches masked visual tokens (Bao et al., 2022), the feature of the masked regions such as HOG (Wei et al., 2021), or reconstruct the masked pixels (He et al., 2021; Xie et al., 2021). Most recently, MGD (Yang et al., 2022b) attempts to generate the entire teacher's feature map by student's masked feature map. In contrast to these approaches, our method operates on *feature level* and aims to *narrow the feature gap between students and teachers*. Besides, our masking ratio is dynamic and knowledge-aware.

## 5    CONCLUSION

In this paper we demonstrate the potential of masked feature prediction in mining richer knowledge from teacher networks. Particularly, we design a prior knowledge-based feature distillation method, named DPK, and use a dynamic mask ratio scheme, achieved by capturing the feature gap between the teacher-student pairs, to dynamically regulate the training process. The extensive experiments show that our knowledge distillation method achieves state-of-the-art performances on the commonly used benchmarks (*i.e.*, CIFAR100, ImageNet, and MS COCO) under various settings. More importantly, our DPK can make the accuracy of students positively correlated with that of teachers. This feature further improves the performance of our method, and provides a 'shortcut' in teacher model selection.

ACKNOWLEDGEMENTS

This work was supported in part by the National Natual Science Foundation of China (NSFC) under Grants No.61932020, 61976038, U1908210 and 61772108. Wanli Ouyang was supported by the Australian Research Council Grant DP200103223, FT210100228, and Australian Medical Research Future Fund MRFAI000085.

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

# A  APPENDIX

## A.1  DATASETS AND METRICS

**CIFAR-100.** CIFAR-100 (Krizhevsky et al., 2009) contains 50K images for training and 10K images for testing, labeled into 100 fine-grained categories. The size of each image is $32 \times 32$. We evaluate the proposed DPK on this dataset with image recognition and report the top-1 accuracy.

**ImageNet.** ImageNet (Deng et al., 2009) consists of 1.2M images for training and 50K images for validation, covering 1,000 categories. All images are resized to $224 \times 224$ during training and testing. We report the top-1 and top-5 accuracy on this dataset for image recognition.

**MS COCO.** MS COCO (Lin et al., 2014) is the most commonly used object detection benchmark, which contains 80 categories. We also conduct experiments for object detection to further evaluate our DPK. In particular, we use `train2017` (118K images) for training, and test on `val2017` (5K images). We adopt the standard evaluation protocol introduced by MS COCO, *e.g.* mAP, $AP_{50}$, $AP_{75}$, *etc*.

## A.2  IMPLEMENTATIONS

**Training details.** On CIFAR-100, the batch size and initial learning rate are set to 64 and 0.05. We train the models for 240 epochs in total with SGD optimizer, and decay the learning rate by 0.1 at 150, 180, and 210 epochs. The weight decay and the momentum are set to 5e-4 and 0.9. On ImageNet, we adopt the SGD optimizer (with 0.9 momentum) to train the student networks for 100 epochs with a batch size of 256. The learning rate is set to 0.1, and we decay it by 0.5 every 25 epochs. We set the weight decay to 0.0001. We also apply the vanilla logits distillation loss (Hinton et al., 2015) in our method. For the loss weights in Eqn. equation 3, we set $\alpha = 0.8$ and $\beta = 0.2$ for all experiments. The temperature $\tau$ used on the ImageNet dataset is set to 1.0, and the same parameter on the CIFAR-100 dataset is set to 4.0. The loss weights of each stage is set to 1.0 in the multi-stage feature distillation setting. Our implementation on MS-COCO for object detection follows the same setting used in (Yang et al., 2022a). We adopt mean squared error (MSE) as the feature distillation loss $\mathcal{D}_{feat}$. All experiments are conducted on 8 Tesla V100 GPUs, and our implementation is based on mmdetection framework (Chen et al., 2019).

**Transformation modules.** To perform feature mimicking, we reshape the original student feature map $\boldsymbol{F}^s \in \mathbb{R}^{\mathsf{C_s} \times \mathsf{H} \times \mathsf{W}}$ into a sequence of flattened 2D patches , where $(\mathsf{H}; \mathsf{W})$ is the resolution of the original student feature map, $C$ denotes the number of channels, $(\mathsf{P}; \mathsf{P})$ is the resolution of each feature patch, and $\mathsf{N} = \mathsf{HW}/\mathsf{P}^2$ is the resulted number of patches, which also serves as the effective input sequence length for the Transformer (Vaswani et al., 2017).

The Transformer uses constant latent vector size $d$ through all of its layers, so we flatten the patches and map to $d$ dimensions with a trainable linear projection. Particularly, we name the output through this projection the *patch embeddings*, similar to ViT (Dosovitskiy et al., 2020). We also add *position embeddings* to the *patch embeddings* to retain positional information. We use standard learnable 1D position embeddings, since no significant performance gain has been observed from using more advanced 2D-aware position embeddings. The resulted sequence of embedding vectors serves as input to the Transformer encoder. The Transformer encoder (Vaswani et al., 2017) consists of alternating layers of multi-head self attention and MLP blocks. Layernorm (LN) is applied before every block, and residual connections after every block. Specifically, the number of encoder layer/block is set to 6 in all experiments.

Same design for the original teacher feature map $\boldsymbol{F}^t \in \mathbb{R}^{\mathsf{C}_t \times \mathsf{H} \times \mathsf{W}}$ is adopted. The output of the student encoder is referred to as the *student tokens* $\mathbf{F}^s \in \mathbb{R}^{\mathsf{N} \times \mathsf{d}}$, and the output of the teacher encoder is referred to as the *teacher tokens* $\mathbf{F}^t \in \mathbb{R}^{\mathsf{N} \times \mathsf{d}}$. Notice that $\mathbf{F}^s$ and $\mathbf{F}^t$ are now dimensionally aligned. Then we can perform masking on them and eventually construct the so-called *hybrid tokens* $\mathbf{F}^h$.

The shared decoder is designed to perform the teacher feature prediction task. We send the *hybrid tokens* to the shared decoder, which finally produces the output tokens of the same dimension as the original teacher feature map $\boldsymbol{F}^t \in \mathbb{R}^{\mathsf{C}_t \times \mathsf{H} \times \mathsf{W}}$. Feature distillation losses are thus computed between the output tokens and the original teacher feature map. During inference, all transformation modules are dropped. Therefore, there is no additional computational cost over the original student network.

- `Conv`: consisting of several convolutional layers, a pooling layer and a fully connected layer.
- `Encoder-Decoder`: We apply ViT Dosovitskiy et al. (2020)-based encoder/decoder in our default transformation module. The encoder Vaswani et al. (2017) consists 6 blocks, and decoder consists of 6 blocks for the single-stage feature distillation and 4 blocks for the multi-stage feature distillation. $\mathbf{F}^s$ and $\mathbf{F}^t$ are encoded by their encoders to form hybrid tokens, and then used as the input of the shared decoder.
- `Decoder`: Different from encoder-decoder, $\mathbf{F}^s$ and $\mathbf{F}^t$ do not need to go through their respective encoder networks, but align their feature dimensions through a layer of simple MLP, then add position encoding to form hybrid tokens as the input of the decoder network.

**Details of ConvNeXt-E.** To evaluate the proposed DPK, we conduct experiments (Fig. 4) on the recently published ConvNeXt (Liu et al., 2022). We take four ConvNeXt variants, ConvNeXt-T/S/B/L, as teachers, and build a build a smaller **ConvNeXt-E** to serve as the student by reducing the blocks/channels in each stage. The other details, such as training strategies, are same as the official models. The configurations are summarized in Table 10. We also report the numbers of parameters, FLOPS, and accuracy for reference.

Table 10: **Detailed settings of ConvNeXt variants.** We also report the parameters, FLOPS and the performance.

|  | Architecture | Channels | Blocks | #Param. | #FLOPS | Top-1 Acc. | Top-5 Acc. |
|---|---|---|---|---|---|---|---|
| Student | ConvNeXt-E | (96,192,384,768) | (2,2,4,2) | 17.43M | 2.6G | 77.43% | 93.30% |
| Teacher | ConvNeXt-T | (96,192,384,768) | (3,3,9,3) | 28.59M | 4.5G | 82.06% | 95.85% |
|  | CovnNeXt-S | (96,192,384,768) | (3,3,27,3) | 50.22M | 8.7G | 83.15% | 96.43% |
|  | ConvNeXt-B | (128,256,512,1024) | (3,3,27,3) | 88.59M | 15.4G | 86.59% | 98.19% |
|  | ConvNeXt-L | (192,384,768,1536) | (3,3,27,3) | 197.77M | 34.4G | 87.40% | 98.37% |

**Details for object detection.** We implement Faster RCNN (Ren et al., 2015) and RetinaNet (Lin et al., 2017b) with different backbones. To integrate the multi-scale features, FPN (Lin et al., 2017a) is adopted for all experiments. We take FGD (Yang et al., 2022a) (recently published on CVPR'22) as the baseline model, and build our method on it. In particular, the feature distillation loss in FGD can be formulated as follows:

$$\mathcal{L}_{feat} = w_f \mathrm{M A^S A^C}(\mathbf{F}^t - f(\mathbf{F}^s))^2 + w_b(1 - \mathrm{M})\mathrm{A^S A^C}(\mathbf{F}^t - f(\mathbf{F}^s))^2, \tag{8}$$

where $\mathbf{F}^s$, $\mathbf{F}^t$ denotes the feature map from student detector and teacher detector, respectively. $f$ is the adaption layer to reshape the $\mathbf{F}^s$ to the same dimension as $\mathbf{F}^t$. M is the binary mask, indicating the foreground and background regions, derived from the ground truth. $\mathrm{A}^S$, $\mathrm{A}^C$ denote the learnable spatial attention map and channel attention map used in FGD. Besides, $w_f$ and $w_b$ are the hyper-parameters to balance the losses for foreground and background regions. To combine our model with FGD, we replace the feature mimicking part with Eq. 4, and we have:

$$\mathcal{L}_{feat} = w_f \mathrm{M A^S A^C} \mathcal{D}_{feat}(\mathbf{F}^t, \mathbf{F}^h) + w_b(1 - \mathrm{M})\mathrm{A^S A^C}\mathcal{D}_{feat}(\mathbf{F}^t, \mathbf{F}^h), \tag{9}$$

where $\mathbf{F}^h$ denotes the *hybrid features*, whose construction process is stated in the main paper. As for the hyper-parameters, such as $w_f$ and $w_b$ in Eq. 9, we follow the settings in FGD and fine-tune them on each training fold. Specifically, we adopt the hyper-parameters $\{w_f = 5\mathrm{E}{-}5, w_b = 2.5\mathrm{E}{-}5\}$ for all the two-stage detectors, and $\{w_f = 2\mathrm{E}{-}3, w_b = 5\mathrm{E}{-}4\}$ for all the one-stage detectors. We train all the detectors for 24 epochs with SGD optimizer, with the momentum as 0.9 and the weight decay as 0.0001.

## A.3    COMPARISON WITH TAKD

TAKD (Mirzadeh et al., 2020) introduces intermediate models as teacher assistants (TAs) to bridge the capacity gap between the teacher models and the student models. This work shares similar motivation with DPK, and we provide additional experiments to compare these two works. In particular, we report the performance of TAKD with one TA (official setting), here we compare our method and

TAKD with more TAs. The experiments are based on the ResNet, and we use ResNet-101 as the teacher and ResNet-18 as the student. For TAKD, we adopt ResNet-50 and ResNet-34 as TAs, and then train the TAs and students one by one. For DPK, we directly train the student under the teacher's guidance. The results are summarized in Table 11. We can find that our DPK surpasses TAKD in performance (*e.g.* 73.00 *v.s.* 71.41). Also, note DPK does not need multiple training.

Table 11: **Comparison of DPK and TAKD on ImageNet.** We use ResNet-18 as the student model. For our DPK, we train the student model under the guidance of ResNet-101. For TAKD, we use ResNet-101 as the teacher model, and use ResNet-50 and ResNet-35 as TAs.

| Acc. | Baseline (R18) | R101→R50→R34→R18 (TAKD) | R101→R18 (DPK) |
|---|---|---|---|
| Top-1 | 69.55 | 71.41 | **73.00** |
| Top-5 | 89.09 | 90.22 | **90.95** |

## A.4 MORE EXPERIMENTS FOR HETEROGENEOUS MODELS

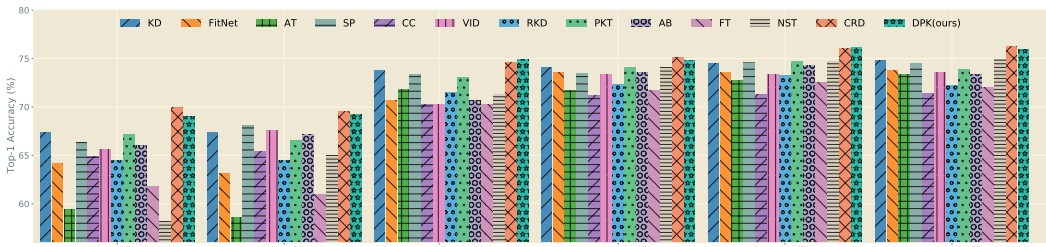

Figure 6: **Heterogeneous experiments on CIFAR-100.** Top-1 accuracy is reported. Best viewed in color with zoom in. "T" denotes the teacher, and "S" denotes the student. Statistically, DPK ranks $1^{st}$ for two pairs and $2^{nd}$ for four pairs.

We report the heterogeneous experiments for some common settings in the main paper. In this section, we give the experiments on CIFAR-100, and more cases on ImageNet.

**Experiments on CIFAR-100.** Fig. 6 presents the experimental results on CIFAR-100. According to these results, we can find our model outperforms all baseline methods across two model pairs, *i.e.* ResNet50(T)-VGG8(S) and WRN40-2(T)-ShuffleNetV1(S), and ranks second for the remaining four model pairs, which demonstrates the effectiveness and robustness of DPK. Besides, we also observe that CRD (Tian et al., 2020) achieves promising performance on this dataset for heterogeneous settings. Note that our method significantly performs better than CRD for homogeneous settings on CIFAR-100 (see Table 1) and ImageNet (see Table 2) and heterogeneous settings on ImageNet (see Table 3).

**Experiments on ImageNet.** On ImageNet, we adopt EfficientNet (Tan & Le, 2019) as the teacher and ResNet18 as the student to conduct heterogeneous experiments. We only distill the features in the last stage for this experiment. The results shown in Table 12 suggest that DPK also works for other teacher-student pairs, and can be further improved by applying larger models.

## A.5 MORE EXPERIMENTS ON TRANSFORMER-STYLE TEACHER-STUDENT DISTILLATION PAIRS

**Performance on ImageNet.** We first validate the effectiveness of the proposed method on different transformer architectural distillation pairs on ImageNet-1K. The results are shown in Table 13. From Table 13, we can see that DPK outperforms all the competitors. To be specificl, DPK obtains 6.78% gain over Hard (Touvron et al., 2021) on DeiT-Tiny (77.1 % v.s. 74.5%), which introduces a hard decision token and a distillation token for distilling the inductive bias from a large pretrained CNN teacher. When applying to the prevalent Swin-tiny (Liu et al., 2021b), a hierarchical vision transformer using shifted windows, DPK can still bring a further 1.7% improvement. The results reval that DPK is not limited to CNN architectures, but also is effective in transformer architectures.

Table 12: **Results for heterogeneous models.** We set ResNet18 as the student and networks from EfficientNet Tan & Le (2019) series as teachers.

| Acc. | Student | EfficientNet-B0 | EfficientNet-B1 | EfficientNet-B2 | EfficientNet-B3 | EfficientNet-B4 |
|---|---|---|---|---|---|---|
| Top-1 | 69.55 | 72.46 | 73.39 | 73.81 | 73.89 | 74.05 |
| Top-5 | 89.09 | 90.56 | 91.10 | 91.56 | 91.63 | 91.58 |

Table 13: **Comparisons of different transformer-based distillation pairs on ImageNet-1K. KD**: the vanilla KD algorithm popularized by Hinton et al. (Hinton et al., 2015). **Hard**: similar to DeiT (Touvron et al., 2021), we introduce a hard decision and a distillation token. **Manifold**: training a tiny student model to match a pre-trained teacher model in the patch-level manifold space (Jia et al., 2021).

| Distillation Method | Teacher | Top-1 Acc. (%) | Student | Top-1 Acc. (%) |
|---|---|---|---|---|
| Vanilla Student | | | | 72.2 |
| KD (Hinton et al., 2015) | | | | 73.0 |
| Hard (Touvron et al., 2021) | | | | 74.5 |
| Manifold (Jia et al., 2021) | CaiT-S24 | 83.4 | DeiT-Tiny | 76.5 |
| **Ours** | | | | **77.1** |
| Vanilla Student | | | | 79.9 |
| KD (Hinton et al., 2015) | | | | 80.0 |
| Hard (Touvron et al., 2021) | | | | 81.3 |
| Manifold (Jia et al., 2021) | CaiT-S24 | 83.4 | DeiT-Small | 82.2 |
| **Ours** | | | | **82.5** |
| Vanilla Student | | | | 81.2 |
| KD (Hinton et al., 2015) | | | | 81.7 |
| Hard (Touvron et al., 2021) | | | | 81.7 |
| Manifold (Jia et al., 2021) | Swin-Small | 83.2 | Swin-Tiny | 82.2 |
| **Ours** | | | | **82.6** |

Table 14: **Distillation results of gradually increasing teacher capacity on ImageNet-1K.**

| | Teacher Models | #Param. | Teacher Acc. Top-1 (%) | Student Acc. Top-1 (%) |
|---|---|---|---|---|
| Student | ViT-Tiny | 5.01M | – | 72.2 |
| Teacher | ViT-small | 22.05M | 79.9 | 73.8 |
| | ViT-base | 86.57M | 81.8 | 74.0 |
| | CaiT-S24 | 46.92M | 83.4 | 76.1 |
| Student | Swin-tiny | 29M | – | 81.2 |
| Teacher | Swin-small | 49.61M | 83.2 | 82.6 |
| | Swin-base | 87.78M | 85.2 | 82.7 |
| | Swin-large | 196.53M | 86.3 | 82.9 |

**Bigger Model, Better Teacher.** Recall that DPK performs well for varying-sized CNN-based teacher-student transfer pairs. We thus conlude that *students can consistently benefit from teachers with higher capacity using DPK*. Here, we see whether this conclusion holds for transformer-style transfer pairs by progressively using larger teachers. Perhaps not surprisingly, as listed in Table 14, the accuracy of the student distilled by our method is continuously improved by progressively replacing larger teacher models, while the students guilded by other algorithms fluctuates in performance, reaffirming our conclusion.

Table 15: **ResNet-18 trained with varying teacher models.** We report the top-1 and top-5 accuracy on ImageNet. All teacher models are re-trained by us to adjust their final performances.

| | Models | #Param. | Teacher Acc. (%) | | Student Acc. (%) | |
|---|---|---|---|---|---|---|
| | | | Top-1 | Top-5 | Top-1 | Top-5 |
| Student | ResNet18 | 11.69M | – | – | 69.55 | 89.09 |
| Teacher | ResNet34_v1 | 21.80M | 73.31 | 91.42 | 70.68 | 90.16 |
| | ResNet34_v2 | 21.80M | 74.37 | 91.90 | 71.36 | 90.08 |
| | ResNet34_v3 | 21.80M | 75.81 | 92.70 | 71.60 | 90.19 |
| | ResNet34_v4 | 21.80M | 76.27 | 92.96 | 71.41 | 90.22 |
| | ResNet34_v3 | 21.80M | 75.81 | 92.70 | 71.60 | 90.19 |
| | ResNet50_v1 | 25.56M | 75.86 | 92.88 | 71.63 | 90.16 |
| | ResNet101_v1 | 44.55M | 75.97 | 92.73 | 71.32 | 89.95 |
| | ResNet152_v1 | 60.19M | 76.24 | 92.80 | 71.21 | 89.89 |

## A.6    FACTORS AFFECTING THE EFFECTIVENESS OF KD

In the main paper, we suppose there are two main factors affecting students' performance: (*i*) the capacity of the teacher model, and (*ii*) the performance of the teacher model. We conduct a *toy experiment* to support these two assumptions in this part.

**Performance of the teacher model.** First, we train ResNet-34 with different teacher models using the vanilla KD (Hinton et al., 2015) to get several ResNet-34 with different performance. Then we use ResNet-18 as student model, ResNet-34 (with different accuracy) as teachers to explore the relation of student's accuracy and teacher's accuracy. As shown in Table 15, we can find better teachers generally lead to better students when the teachers share the same CNN model.

**Capacity of the teacher model.** Similarly, we also select some different distilled ResNet models and keep their performance at a similar level to explore the impact of capacity difference on student's final accuracy. The results in Table 15 suggest that larger teachers generally degrade the students when the teachers have similar performance.

The above two factors make the choice of the teacher models become a special 'trade-off' between accuracy and capacity, and our DPK alleviates this issue by reducing the capacity gap in feature-level.

## A.7    REPRESENTATION TRANSFERABILITY.

Following previous works in (Zhao et al., 2022; Tian et al., 2020), we evaluate the generalization ability of learned representations by transferring them to unseen datasets. Specifically, we adopt a WRN-16-2 student distilled from a WRN-40-2 teacher as a frozen representation extractor (layers before logits) trained on CIFAR 100. We then train a single linear layer classifier on top of the frozen representations perform 10-way (for **STL-10** Coates et al. (2011)) and 200-way (for **TinyImageNet**[2]) classification. To better quantify the transferability of the representations, we keep the representations fixed and only update the linear probing heads. Table 16 compares several baseline methods. From Table 16, we note that, when transferring the representations learned from CIFAR-100 to STL-10 and TinyImageNet, our method outperforms all baselines, confirming its superiority in improving the transferability of representations.

Table 16: Comparison of Top-1 accuracy of different distillation methods on transferring representations learned from CIFAR-100 to STL-10 or TinyImageNet (TinyIN), where WRN-16-2 is taken as the student and WRN-40-2 as the teacher. 'Baseline' means that no distillation method is applied and the student WRN-16-2 is trained from scratch on CIFAR-100.

| | Baseline | KD | AT | FitNet | CRD | CRD+KD | ReviewKD | DKD | Ours |
|---|---|---|---|---|---|---|---|---|---|
| STL-10 | 69.7 | 70.9 | 70.7 | 70.3 | 71.6 | 72.2 | 72.4 | 72.9 | 73.2 |
| TinyIN | 33.7 | 33.9 | 34.2 | 33.5 | 35.6 | 35.5 | 36.6 | 37.1 | 37.8 |

---

[2]https://www.kaggle.com/c/tiny-imagenet

### A.8 Visualizations

**CKA curve in training.** To qualitatively analyze the proposed DPK, we take ResNet-18 as the student model, and visualize the CKA similarities trained with four teacher models in the training phase. As shown in Fig. 7, we can find the following two observations: (*i*) the CKA values increase with training, which demonstrates that our DPK does narrow the gap of teacher-student models at feature level, and (*ii*) the CKA for the lager teacher is significantly lower than that for small teacher, which suggests the necessity of our dynamic design (observation (*i*) also support this conclusion). We also visualize the dynamic mask ratios in Fig. 8 for reference.

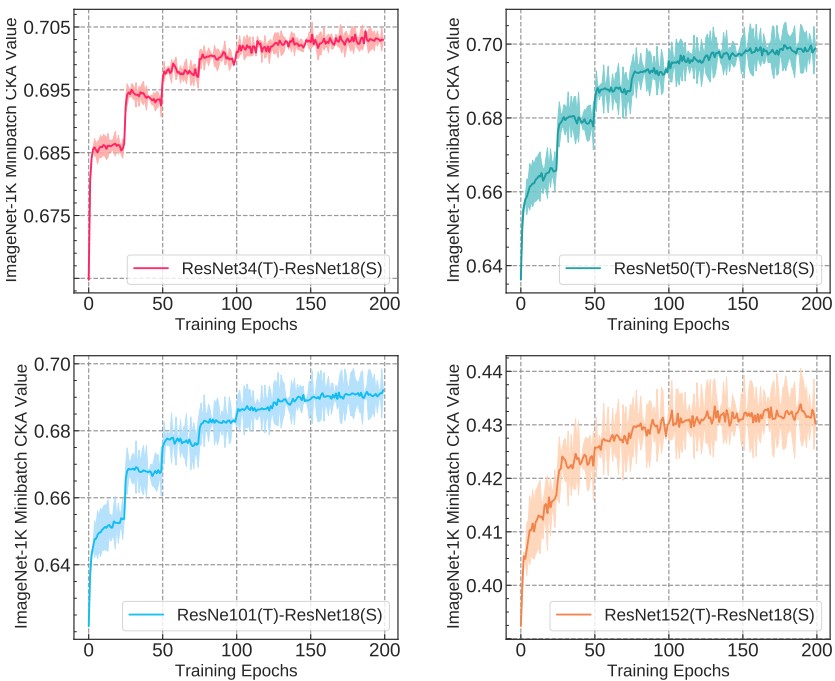

Figure 7: **CKA curves in the training phase.** We visualize the CKA similarities in the training for four teacher-student pairs. The CKA values are computed at the batch-level, and we average them at the epoch-level for better presentation. The corresponding mask ratios to these CKA values are visualized in Fig. 8.

**Feature similarity before/after distillation.** Here we give more visualizations to show the feature similarity between students and teachers before/after distillation. The feature similarities measured by CKA are shown in Fig. 9 and the features similarities measured by Cosine are shown in Fig. 10. These results qualitatively show the effectiveness of our DPK.

### A.9 Complexity analysis,

Here we provide the complexity analysis in Table 17, which may be helpful to the potential users of the proposed model. Particularly, we report the parameters, FLOPS, and corresponding performance (top-1 accuracy on ImageNet) for some baseline models and several variants of DPK. The parameters and FLOPS are only counted for the transformation modules, and the corresponding numbers for ResNet-34 are 21.8M and 3.6G respectively. As shown in Table 17, we can see that it is optional to use several convolution layers or a lightweight decoder to realize teacher/student feature transformation. Furthermore, compared with some previous feature-based KD methods such as OFD (Heo et al., 2019a), TOFD (Zhang et al., 2020), VID (Ahn et al., 2019) and ReviewKD (Chen et al., 2021), DPK has a comparable computational overhead but always shows superior distillation results when using the same feature transformation. From these results, we can also find that a lightweight ViT-style encoder-decoder (1-1) can achieve a top-1 accuracy of 72.43, which surpasses all counterparts (including those with heavier transformation modules such as ReviewKD). These results also suggest

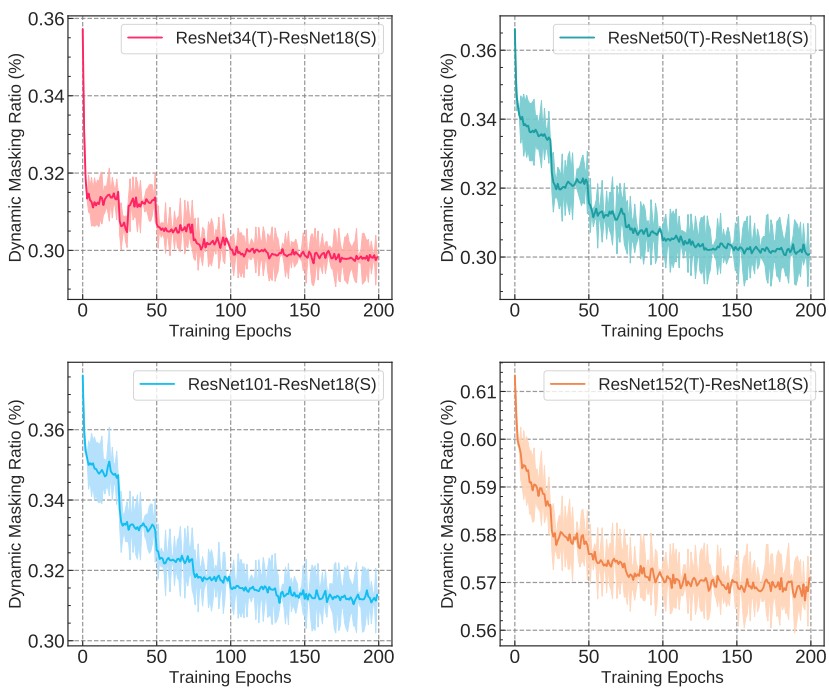

Figure 8: **Dynamic mask ratios.** We visualize the mask ratios dynamically adjusted according to CKA. The mask ratios are adjusted at the batch-level, and we average them at the epoch-level for better presentation. The corresponding CKA curves are visualized in Fig. 7

the performance can be further improved with stronger transformation modules, and the potential users can choose different modules accordingly.

| Methods | Teacher Transform | Student Transform | Top-1 Acc. | #params | FLOPs |
|---------|-------------------|-------------------|------------|---------|-------|
| FitNet (Romero et al., 2015) | None | 1x1 conv | 70.81 | 5M | 0.9G |
| AT (Zagoruyko & Komodakis, 2017) | Attention | Attention | 70.63 | 9.3M | 1.7G |
| AB (Heo et al., 2019b) | Binarization | 1x1 conv | 70.76 | 6.1M | 1.4G |
| OFD (Heo et al., 2019a) | Margin ReLU | 1x1 conv | 71.34 | 13.3M | 2.5G |
| TOFD (Zhang et al., 2020) | 1x1 conv | 1x1 conv | 70.29 | 22.9M | 3.2G |
| ReviewKD (Chen et al., 2021) | 1x1 conv | 1x1 conv | 71.61 | 24.2M | 2.8G |
| VID (Ahn et al., 2019) | 4x4 transposed conv | 1x1 conv | 70.43 | 18.8M | 2.2G |
| DPK | 1x1 conv | 1x1 conv | 71.39 | 17.2M | 2.4G |
| DPK | None | MLP-Decoder | 72.37 | 18.3M | 2.5G |
| DPK | None | Enc.-Dec. (1-1) | 72.43 | 10.5M | 1.6G |
| DPK | None | Enc.-Dec. (2-2) | 72.31 | 20.0M | 2.7G |
| DPK | None | Enc.-Dec. (4-2) | 72.46 | 32.6M | 4.5G |
| DPK | None | Enc.-Dec. (4-4) | 72.48 | 38.9M | 4.8G |
| DPK | None | Enc.-Dec. (6-4) | 72.46 | 51.5M | 5.9G |

Table 17: Settings: ResNet34 as student and ResNet18 as student. All experiments are conducted on eight Tesla V100 GPUs using the same image augmentations, batch size. The symbols $(i - j)$ indicates that the encoder contains $i$ layers and the decoder contains $j$ layers.

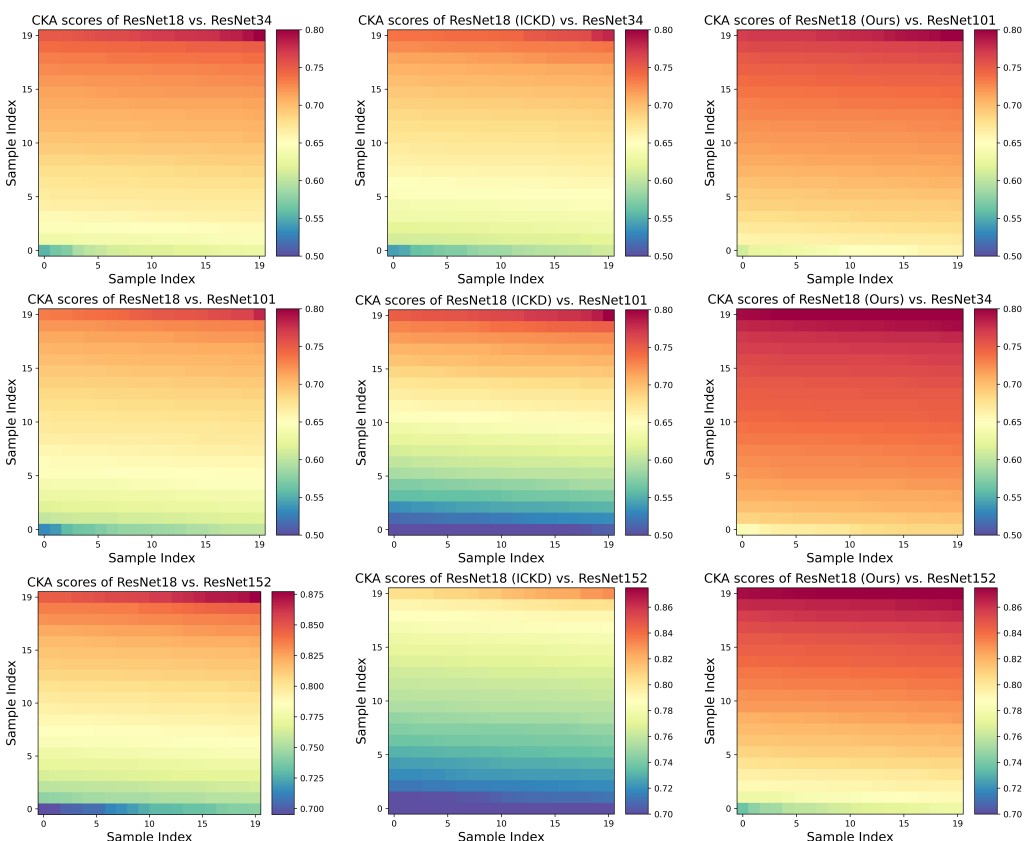

Figure 9: **Feature similarity measured by CKA.** We visualize the CKA similarities for teacher-student pairs before/after distillation. We adopt the same setting used in Fig. 5 for better presentation.

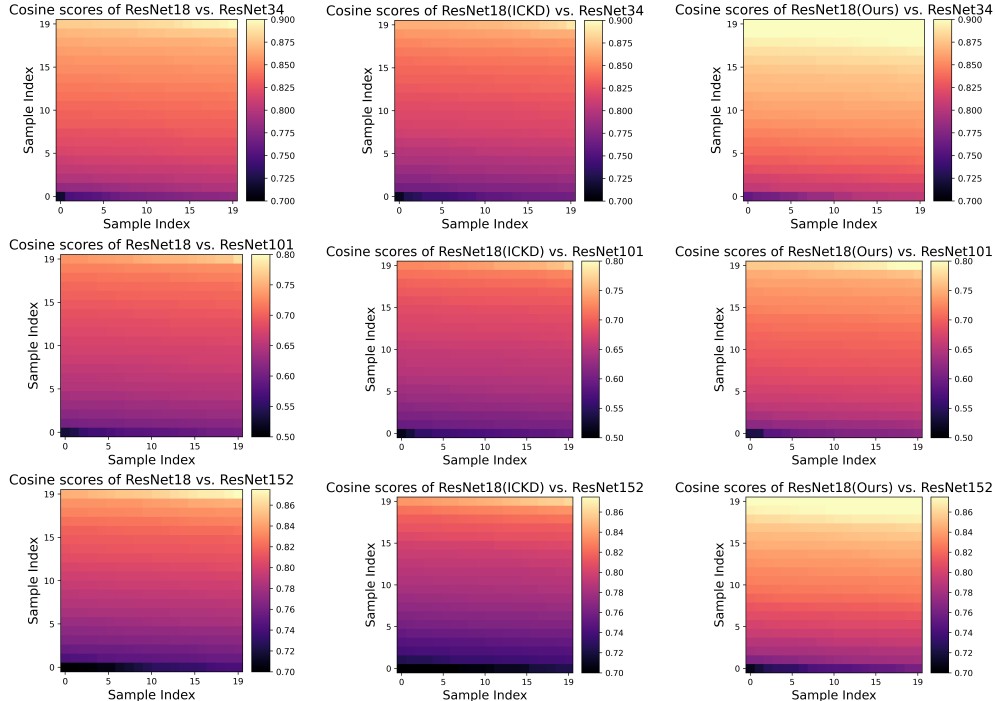

Figure 10: **Feature similarity measured by Cosine distance.** We visualize the cosine similarities for teacher-student pairs before/after distillation. We adopt the same setting used in Fig. 5 for better presentation.

## A.10   LIMITATIONS AND DISCUSSIONS

**Limitations.** The Fig. 1, 3, 4 and Table 4, 12 report that the proposed DPK can continue to benefit from larger teacher models. However, we also observe some outliers. In particular, when the teachers of different size belong to different model architectures, the student trained with best teacher may not perform best. For example, the ResNet-18 trained with ConvNeXt-T achieves 71.80 top-1 accuracy on ImageNet (see Figure 4), and then it can achieve 72.51 top-1 accuracy under the guidance of ResNet-34 (see Table 2). Although ConvNeXt-T performs better than ResNet-34, but it provides less guidance to the students. Note that this does not always happen, and we argue that the better teachers still give better results when they belong to the same network family.

**Future work.** This paper investigates a new paradigm for feature distillation, *i.e.* introducing the teacher's feature to the student as prior knowledge before conducting feature distillation. We show the effectiveness of this idea, meanwhile, this idea can be further explored in future work. For example, we randomly mask the student's features and fill them with the teacher's feature. A possible solution is to actively choose the prior knowledge according to some rules, such as the discriminability (Zhou et al., 2016) or uncertainty (Kendall & Gal, 2017). Furthermore, different tasks may require different kinds of prior knowledge, *e.g.* object detection may focus more on the foreground features than background features. We hope the idea of this work and the relevant discussions can provide insights to the community.

## A.11   POTENTIAL IMPACTS.

DPK aims to learn more powerful representations for the student model, and theoretically, it can be applied to most CNN-based models and tasks, including these may have negative impacts on the society (*e.g.* face recognition). Besides, same as other data-driven methods, DPK may also give biased results if the models are trained from biased data.

