# OpenReview forum: "Better Teacher Better Student: Dynamic Prior Knowledge for Knowledge Distillation"
_ICLR.cc/2023/Conference — ICLR 2023 poster_

### Official Review · Reviewer_aN37 · 2022-10-21

**Confidence:** 4
**Correctness:** 4
**Technical Novelty And Significance:** 3
**Empirical Novelty And Significance:** 3
**Recommendation:** 8

**Clarity, Quality, Novelty And Reproducibility:**

The paper is well-written and shows justifying ablation study results that support the proposed idea of this work. Also, it shows SOTA performance compared to existing methods. Its overall clarity, quality, and novelty are good enough, and it provides detailed information to reproduce the reported results.


**Strength And Weaknesses:**

Strength
- To the author’s best of knowledge, this work is the first work to take the features of teachers as ‘input’, not just ‘target’ in knowledge distillation. If this is true, this paper can take credit for novelty. Also, this work showed the ‘large models are not always better teachers’ issue and tried to solve the issue with the proposed idea. This work did a good job of analyzing the introduced problem with an experiment. The proposed method shows the SOTA performance in various benchmarks regarding the classification and the object detection task.

Weakness
- It seems that the transformation module is too heavy, for example, it uses the ViT encoder and decoder module as the feature transformation module which requires heavy computation and process. Also, its dynamic mechanism process is a little hard to follow.
In table 8, the Encoder-Decoder shows only marginal performance gain compared to MLP-Decoder, considering the heavy computation required by the ViT encoder.

- Since the encoder-decoder is ViT, it would have been more informative to address the training computational cost, even though it does not affect the test phase.




**Summary Of The Paper:**

This paper proposes to use dynamic prior knowledge (DPK) that uses the teacher’s feature as inputs for the feature distillation by mixing the student features and using an encoder-decoder framework that learns the teacher’s feature as the target. Using DPK has an advantage in that the performance of the student network has a positive correlation with that of the teacher.
The performance gain in image classification datasets (CIFAR-100, ImageNet) and object detection (COCO) is remarkable, achieving SOTA performance.


**Summary Of The Review:**

I recommend this paper to be accepted since it tackles a very important issue in knowledge distillation and solved the issue with a novel idea. It has done extensive experiments to support the proposed idea and the results.  However, its performance gain shown in the ablation study is somewhat marginal.

---

> ### Author Response · Authors · 2022-11-15
> **Response to Reviewer aN37**
>
> Thanks for your valuable comments and appreciation for this work. Here we mainly answer the weaknesses in the rebuttal period.
>
> > W1.1: It seems that the transformation module is too heavy ...
>
> First, we agree that using the vanilla ViT encoder-decoder requires heavy computation. Note, however, that the ViT-based transformation module is optional for DPK and we also provide other choices in Table 8 of the main paper. For example, we can use serval convolution layers or a lightweight MLP decoder to implement the feature transformation module. Besides, we also provide the complexity analysis in the following table, which may be helpful to the potential users of the proposed model. Particularly, we report the #params, FLOPS, and corresponding performance (top-1 accuracy on ImageNet) for some baseline models and several variants of DPK in the following table. The #params and FLOPS are only for the transformation modules, and the corresponding numbers for ResNet-34 are 21.8M and 3.6G respectively. From these results, we can find that a lightweight ViT-style encoder-decoder (1-1) can achieve a top-1 acc. of 72.43, which surpasses all counterparts (including those with heavier transformation modules such as ReviewKD). These results also suggest the performance can be further improved with stronger transformation modules, and the potential users can choose different modules accordingly.
>
> | Methods  | Teacher Transform   | Student Transform | Top-1 Acc. | #params | FLOPs |
> | -------- | ------------------- | ----------------- | ---------- | ------- | ----- |
> | FitNet   | None                | 1x1 conv          | 70.81      | 5M      | 0.9G  |
> | AT       | Attention           | Attention         | 70.63      | 9.3M    | 1.7G  |
> | AB       | Binarization        | 1x1 conv          | 70.76      | 6.1M    | 1.4G  |
> | OFD      | Margin ReLU         | 1x1 conv          | 71.34      | 13.3M   | 2.5G  |
> | TOFD     | 1x1 conv            | 1x1 conv          | 70.29      | 22.9M   | 3.2G  |
> | ReviewKD | 1x1 conv            | 1x1 conv          | 71.61      | 24.2M   | 2.8G  |
> | VID      | 4x4 transposed conv | 1x1 conv          | 70.43      | 18.8M   | 2.2G  |
> | DPK      | 1x1 conv            | 1x1 conv          | 71.39      | 17.2M   | 2.4G  |
> | DPK      | None                | MLP-Decoder       | 72.37      | 18.3M   | 2.5G  |
> | DPK      | None                | Enc.-Dec. (1-1)   | 72.43      | 10.5M   | 1.6G  |
> | DPK      | None                | Enc.-Dec. (2-2)   | 72.31      | 20.0M   | 2.7G  |
> | DPK      | None                | Enc.-Dec. (4-2)   | 72.46      | 32.6M   | 4.5G  |
> | DPK      | None                | Enc.-Dec. (4-4)   | 72.48      | 38.9M   | 4.8G  |
> | DPK      | None                | Enc.-Dec. (6-4)   | 72.46      | 51.5M   | 5.9G  |
>
>
>
> > W1.2: .. its dynamic mechanism process is a little hard to follow ...
>
> Thanks for this feedback, and we will revise the related texts to make it easier to understand.
>
>
> > W2: it would have been more informative to address the training computational cost, even though it does not affect the test phase.
>
> Thanks for this suggestion. We agree that it would be useful if the computational cost can be further reduced, even though it does not affect the inference. In the follow-up works, we will explore some lightweight and more efficient architectures for teacher and student feature transformation.

---

### Official Review · Reviewer_cqrC · 2022-10-23

**Confidence:** 5
**Correctness:** 2
**Technical Novelty And Significance:** 2
**Empirical Novelty And Significance:** 3
**Recommendation:** 5

**Clarity, Quality, Novelty And Reproducibility:**

This paper presents a dynamic prior knowledge mechanism for feature distillation to provide a solution to the ‘larger models are not always better teachers’ issue. The proposed method is simple and technically sound. However, the motivation behind "why prior knowledge can relieve the feature gap" is unclear.

**Strength And Weaknesses:**

Main Review:

Strength:

1)This paper presents a dynamic prior knowledge mechanism for feature distillation to provide a solution to the ‘larger models are not always better teachers’ issue. The proposed method is simple and technically sound.

2)The experimental results of the proposed method outperform that of the other comparable distillation knowledge models.

3)This paper is well-written and easy to follow.

Weaknesses:

1)The technical novelty of the proposed method is somewhat limited, and the authors did not provide theoretical analysis to support the proposed algorithm. This article lacks a strong reason to explain why the proposed prior knowledge mechanism can perform well. What is the relationship between the proposed prior knowledge mechanism and the feature gap?

2)There is no detailed description of the progress of knowledge distillation research in the introduction. Are there any other articles that attempt to address the ‘larger models are not always better teachers’ issue? I think this article lacks the most relevant baseline, e.g. [1].

3)Is the prior knowledge mechanism that incorporates the teacher's feature into the student model fair in comparison to other baselines?

4)This paper lacks complexity analysis, which is important.

[1] Beyer L, Zhai X, Royer A, et al. Knowledge distillation: A good teacher is patient and consistent[C]//Proceedings of the IEEE/CVF Conference on Computer Vision and Pattern Recognition. 2022: 10925-10934.


**Summary Of The Paper:**

This paper proposes the prior knowledge mechanism for feature distillation, which can fully excavate the distillation potential of big models. Furthermore, the authors propose the dynamic prior knowledge (DPK) to solve the ‘larger models are not always better teachers’ issue, which makes the performance of the student model positively correlated with that of the teacher model.
Finally, experiments are conducted to verify the effectiveness of the proposed algorithm.


**Summary Of The Review:**

It lacks evidence and reasons to support its assumption on prior knowledge, and also this assumption may be unfair to the general teaching. Besides, the authors avoided some related work from machine teaching. I don't know why.

---

> ### Author Response · Authors · 2022-11-15
> **Response to Reviewer cqrC (Part I)**
>
> We thank all the valuable comments from Reviewer cqrW for this work. Here we mainly answer the weaknesses in the rebuttal period.
>
> > W1.1: The technical novelty of the proposed method is somewhat limited ...
>
> This work does adopt some existing techniques, while it is totally different from other works at the conceptual level, which is the main contribution of this work. Specifically, masked image modeling (MIM) is mainly used to obtain well-generalized and scalable representations by self-supervised learners. While this work introduces MIM into knowledge distillation (KD) with a dynamic masking strategy to bridge the capacity gap between teachers and students. Compared with other KD methods, this work is the first to introduce the teacher's features as the 'input' in feature-based KD, not just the ‘target’. As a result, our model achieves better results and addresses the ‘larger models are not always better teachers’ issue. Therefore, we argue that the proposed method can provide insights to the community.
>
>
> > W1.2: This article lacks a strong reason to explain why the proposed prior knowledge mechanism can perform well.
>
> Intuitively, better teachers should give better supervision for the students, thus the students should get better performance. However, the experiments give contradictory results. In this work, we find this issue is mainly caused by the large capacity gap between students and teachers (see Appendix A.6 for corresponding experiments). To this end, the proposed method aims to reduce the capacity gap by integrating the features of teachers and students before applying knowledge distillation. The visualizations in Figure.5 (in the main paper),  Figure.9 (in Appendix), and Figure.10 (in Appendix) confirm that the students trained with DPK output more similar features with the teachers than other baselines. Besides, we also give more instance-level visualizations in this anonymous link (https://anonymous.4open.science/r/dpk_3440/README1.md) to illustrate the proposed method.
>
>
> >  W1.3: What is the relationship between the proposed prior knowledge mechanism and the feature gap?
>
> The bigger models generally have larger capacities and then can learn more discriminative features. Due to the large capacity gap, the students can not fully learn the features of the larger teachers. The proposed DPK can provide prior knowledge (represented by the features) from teachers to the students before applying knowledge distillation, thereby reducing the feature gap.
>
>
> > W2.1: There is no detailed description of the progress of knowledge distillation research in the introduction.
>
> We provide a brief description of related work in the introduction section, and more reviews for the linterature are given in the related work section (Section 4).
>
> > W2.2 Are there any other articles that attempt to address the ‘larger models are not always better teachers’ issue?
>
> Except for this work, [1-4] also report this issue, while they did not provide a solution. We point out this in the introduction and related work sections.
>
> > W2.3: I think this article lacks the most relevant baseline, e.g. [1].
>
> Thanks for this suggestion. We will revise the related work to cover more related works. However, this work [5] is not suitable for comparison as the baseline model, because it aims to investigate the ultimate performance of the distilled ResNet-50 under a very long training schedule and consistency strategy. This work trains the ResNet-50 on ImageNet for 9600 epochs, almost 100x longer than the standard setting. So applying this method as the baseline model is unfair. Considering this work did not propose novel designs and only apply the logits-based distillation (with the consistent training strategy and very long schedule), the accuracy of classic logit-based KD [1] can be regarded as the result of [5] with a standard training schedule.
>
> > W3: Is the prior knowledge mechanism that incorporates the teacher's feature into the student model fair in comparison to other baselines?
>
> We are convinced that the comparison is fair. Particularly, in the training phase, both the proposed model and other counterparts use pre-trained teachers to provide guidance, and only the students are used in the inference phase.

---

> ### Author Response · Authors · 2022-11-15
> **Response to Reviewer cqrC (Part II)**
>
> > W4: This paper lacks complexity analysis, which is important.
>
> Here we provide the complexity analysis in the following table, which may be helpful to the potential users of the proposed model. Particularly, we report the #params, FLOPS, and corresponding performance (top-1 accuracy on ImageNet) for some baseline models and several variants of DPK in the following table. The #params and FLOPS are only for the transformation modules, and the corresponding numbers for ResNet-34 are 21.8M and 3.6G respectively. From these results, we can find that a lightweight ViT-style encoder-decoder (1-1) can achieve a top-1 acc. of 72.43, which surpasses all counterparts (including those with heavier transformation modules such as ReviewKD). These results also suggest the performance can be further improved with stronger transformation modules, and the potential users can choose different modules accordingly.
>
> | Methods  | Teacher Transform   | Student Transform | Top-1 Acc. | #params | FLOPs |
> | -------- | ------------------- | ----------------- | ---------- | ------- | ----- |
> | FitNet   | None                | 1x1 conv          | 70.81      | 5M      | 0.9G  |
> | AT       | Attention           | Attention         | 70.63      | 9.3M    | 1.7G  |
> | AB       | Binarization        | 1x1 conv          | 70.76      | 6.1M    | 1.4G  |
> | OFD      | Margin ReLU         | 1x1 conv          | 71.34      | 13.3M   | 2.5G  |
> | TOFD     | 1x1 conv            | 1x1 conv          | 70.29      | 22.9M   | 3.2G  |
> | ReviewKD | 1x1 conv            | 1x1 conv          | 71.61      | 24.2M   | 2.8G  |
> | VID      | 4x4 transposed conv | 1x1 conv          | 70.43      | 18.8M   | 2.2G  |
> | DPK      | 1x1 conv            | 1x1 conv          | 71.39      | 17.2M   | 2.4G  |
> | DPK      | None                | MLP-Decoder       | 72.37      | 18.3M   | 2.5G  |
> | DPK      | None                | Enc.-Dec. (1-1)   | 72.43      | 10.5M   | 1.6G  |
> | DPK      | None                | Enc.-Dec. (2-2)   | 72.31      | 20.0M   | 2.7G  |
> | DPK      | None                | Enc.-Dec. (4-2)   | 72.46      | 32.6M   | 4.5G  |
> | DPK      | None                | Enc.-Dec. (4-4)   | 72.48      | 38.9M   | 4.8G  |
> | DPK      | None                | Enc.-Dec. (6-4)   | 72.46      | 51.5M   | 5.9G  |
>
>
>
> References:
>
> [1] On the efficacy of knowledge distillation, Jang Hyun Cho and Bharath Hariharan, ICCV'19
>
> [2] Improved knowledge distillation via teacher assistant, Seyed Iman Mirzadeh, et. al., AAAI'20
>
> [3] Distilling the knowledge in a neural network. Geoffrey Hinton, et. al., 2015
>
> [4] Exploring inter-channel correlation for diversity-preserved knowledge distillation, Li Liu et. al., ICCV'21
>
> [5] Knowledge distillation: A good teacher is patient and consistent, Lucas Beyer, et. al., CVPR'22

---

### Official Review · Reviewer_ERUi · 2022-10-24

**Confidence:** 3
**Clarity, Quality, Novelty And Reproducibility:** See strength and weakness.
**Correctness:** 4
**Technical Novelty And Significance:** 2
**Empirical Novelty And Significance:** 3
**Recommendation:** 6

**Strength And Weaknesses:**

Strength:

1. The authors utilize a transformer-like module to integrate the student and teacher features for feature alignment.
2. Considering the capacity gaps between teacher and student networks, the authors propose a dynamic mechanism for training.
3. The authors do lots of experiments and the proposed method obtain good performance on image classification and objection detection tasks.
4. The paper is well-organized and easy to follow.

Weakness:

1. The technical novelty is marginally novel for the community. Although lots of ablation studies and comparisons demonstrate the good performance of the proposed method, this paper is an incremental work. The authors introduce widely used transformer module and mask ratio for feature distillations.
2. The authors should give more analysis of the Receptive Field (RF) for feature distillation. For object detection, how to facilitate the different RF between teacher and student features is important for feature distillation.
3. In Appendix 4, I notice that CRD outperforms the proposed DPK on four settings. Is the performance also influenced by the RF?
4. I am also curious about the values of dynamic mask ratios for each stage in training. According to the difference between receptive filed, the mask ratio for the shallower stage should be larger than the deeper stage.
5. The similar work [1] should be discussed and compared in the paper. In [1], they use nonlocal module, which is similar to transformer, for feature distillation of object detection. Although the teacher backbone is a little different, they obtain better performance (e.g. Faster RCNN R50(S), 41.5[1] vs 40.6).


[1] Zhang, Linfeng, and Kaisheng Ma. "Improve object detection with feature-based knowledge distillation: Towards accurate and efficient detectors." In International Conference on Learning Representations. 2021.

**Summary Of The Paper:**

The authors propose a new method on feature distillation. They utilize a transformer-like module for feature alignment between each stage of the teacher and student networks. Considering the gaps between teacher and student network, they propose the mask ratios to dynamically guide the feature distillation training. The experiments obtain good performance on image classification (CIFAR-10 and ImageNet) and object detection (COCO).

**Summary Of The Review:**

See strength and weakness.

This is an incremental work for community. The authors introduce some existing techniques into feature distillation. They do lots of exmperiments to demonstrate the superiorities of the proposed DPK method, but lack detailed analyses of the essential problems for feature distillation.

I am glad to improve the final rating if the authors address my concerns or point out that I misunderstand  some parts of this work.

---

> ### Author Response · Authors · 2022-11-15
> **Response to Reviewer ERUi (Part I)**
>
> We sincerely appreciate your thoughtful reviews and hope that our response can address your concerns.
>
> > W1: The technical novelty is marginally novel for the community ...
>
> This work does adopt some existing techniques, while it is totally different from other works at the conceptual level, which is the main contribution of this work. Specifically, masked image modeling (MIM) is mainly used to obtain well-generalized and scalable representations by self-supervised learners. While this work introduces MIM into knowledge distillation (KD) with a dynamic masking strategy to bridge the capacity gap between teachers and students. Compared with other KD methods, this work is the first to introduce the teacher's features as the 'input' in feature-based KD, not just the ‘target’. As a result, our model achieves better results and addresses the ‘larger models are not always better teachers’ issue. Therefore, we argue that the proposed method can provide insights to the community.
>
> By the way, we also provide additional instance-level visualizations in this anonymous link (https://anonymous.4open.science/r/dpk_3440/README1.md) to show that the effectiveness of DPK (students trained with DPK generate more similar features with the teachers than other baselines).
>
> >  W2: ... more analysis of the Receptive Field (RF) for feature distillation ...
>
> Thanks for this suggestion, and here we provide more experiments and discussions. Theoretically, ViT (the default transformation module in DPK) updates the features based on global attention, thus its receptive field is all the features. In contrast, the CNN features have different receptive fields at different feature levels. However, it is difficult to ablate the receptive field without modifying the backbones. To this end, we apply the DPK on different resolution levels to simulate related settings (this is a rough setting for related experiments, and we are glad to conduct extra experiments if the reviewer can provide better experimental settings). Specifically, we evaluate DPK on MS-COCO for the object detection task. The results shown in the following table reveal applying DPK on the deeper features (which have larger receptive fields) gets better overall performance. Besides, also note that distillation on shallower feature layers works better for smaller targets, and distillation on deeper features works better for larger targets. This is mainly because the shallower layers correspond to a smaller receptive field and emphasize the features of small objects.
>
> | Methods | mAP  | AP50 | AP75 | AP-${s}$ | AP-$m$ | AP-${l}$ |
> | ------- | ---- | ---- | ----- | -------- | ------ | -------- |
> |FasterRCNN-101 (T)| 39.9 | 60.1 | 43.3 | 23.5 | 44.2 | 51.5 |
> |Faster-rcnn-50 (S) |38. 4| 59. 0| 42. 0| 21. 5 | 42. 1| 50. 3|
> |+ DPK on the 1-st feature level      |38. 7 | 59. 3| 42. 1| 22. 5| 42. 4 | 51. 3|
> |+ DPK on the 2-nd feature level      |38. 8 |59. 4 | 42. 3| 22. 3| 42. 1 | 51. 5|
> |+ DPK on the 3-rd feature level      |39. 0 |59. 5 | 42. 4 |21. 9 |42. 6 | 51. 4 |
> |+ DPK on the 4-th feature level      |39. 1 |59. 9 |42. 8 |22. 4 |42. 3 | 51. 7|
> |+ DPK on all  feature levels   |40. 6 |60. 7 |44. 4|22. 8| 44. 7 | 54. 0|
> |Retinanet-101 (T)| 38. 9 |58. 0 |41. 5 |21. 0 |42. 8| 52. 4|
> |Retinanet-50 (S) |37. 4| 56. 7| 39. 6| 20. 6|40. 7| 49. 7|
> |+ DPK on the 1-st feature level     |38. 1 | 57. 3 |40. 9 | 21. 4 |41. 6 |50. 5|
> |+ DPK on the 2-nd feature level     |38. 0 |57. 3 |40. 5 |20. 9| 41. 7 | 50. 3|
> |+ DPK on the 3-rd feature level     |38. 2 |57. 4|40. 6 |20. 8| 41. 7| 51. 0|
> |+ DPK on the 4-th feature level     |38. 4|57. 6|41. 0 |21. 5 |41. 7 |51. 3|
> |+ DPK on all feature levels  |39. 7|58. 6|42. 5 |22. 8 |43. 6|53. 6|
>
> > W3: ... CRD outperforms the proposed DPK on four settings. Is the performance also influenced by the RF?
>
> CRD applies contrastive learning with a memory bank to optimize its model. This kind of optimization manner may be more friendly to the small dataset. Considering that the CRD cannot get a good performance on the ImageNet, and our DPK has a larger receptive field (due to the existence of the non-local ViT module),  we tend to think that this is more likely related to the scale of the dataset and the size of images.

---

> ### Author Response · Authors · 2022-11-15
> **Response to Reviewer ERUi (Part II)**
>
> > W4: ... the values of dynamic mask ratios for each stage in training ...
>
> Thanks for your insightful comments. To shed light on how the dynamic masking ratio varies with different distillation stages, we show the change of dynamic masking ratios during training on ImageNet.  Please see additional visualization on this anonymous link: https://anonymous.4open.science/r/dpk_3440/README2.md for details. From the results shown in the figures, we can see that:  i) as you guessed, the overall masking ratio in the shallow layers is larger than that in the deep layers; ii) in the training phase, the masking ratio decreases and converges to a certain range for both the shallow and deep layers.
>
>
> > W5: The similar work [1] should be discussed and compared in the paper ...
>
> Thanks for your suggestion. We are pleased to include a discussion of the connection and difference between DPK and [1] in the revised version. In particular, [1] has two attractive properties: i) attention-guided distillation, letting students’ learning focus on the foreground objects and suppresses students’ learning on the background pixels; and ii) non-local distillation, transferring the relation between different pixels from teachers to students. In summary, both the proposed method and [1] apply the masking strategy (random mask and attention-guided mask) and non-local relation modeling module (transformer and a self-designed non-local module). The key difference is that the proposed model integrates the features of students and teachers with a dynamic mechanism, which is the main contribution of this work. We also note that [1] adopt stronger backbones and then get better performance. For example, it uses Cascade Mask RCNN with ResNeXt101 backbone as the teacher for all the two-stage students and uses RetinaNet with ResNeXt101 backbone as the teacher for all the one-stage students. Besides, these two methods are orthogonal and can be applied together.
>
> References:
>
> [1] Zhang, Linfeng, and Kaisheng Ma. "Improve object detection with feature-based knowledge distillation: Towards accurate and efficient detectors."ICLR. 2021.

---

> > ### Comment · Reviewer_ERUi · 2022-12-08
> > **Final rating after reading rebuttal**
> >
> > The authors give lots of experiments and analyses for DPK and address my concerns, therefore, I decide to improve the final rating.

---

### Decision · Program_Chairs · 2023-01-20

**Decision:**

Accept: poster

**Justification For Why Not Higher Score:**

The reviewers raise the concerns on the limited novelty, comparison with related works and the marginal performance improvement in the experiments. AC reads these comments, the author's response and the paper, and agree with these comments. The performance improvement seems to be more from the ViT encoder-decoder. The idea of masking features in KD has been explored by prior works.

**Justification For Why Not Lower Score:**

This paper investigates an important problem when applying KD in practice. The proposed method is reasonable and inspiring for the community. Most of the reviewers also agree with accepting this paper.

**Metareview: Summary, Strengths And Weaknesses:**

Summary

This paper studies the KD problem when the teacher model is much stronger (i.e., having larger capacity) than the student model and aims to address the model capacity gap issue. To this end, it proposes a dynamic prior knowledge (DPK) method to address this issue. Specifically, the DPK method proposes to replace some features at certain spatial location by the features output from the teachers, in order to relieve the KD-based learning difficulty for the student model.

Strength:

- This paper studies an important problem in KD field, i.e. how to address the capacity gap issue between the teacher and student models.
- The proposed method, DPK, is new in terms to using the teacher model's features as "prior knowledge".

Weakness:
- The overall pipeline is complicated.
- The ViT encoder-decoder seems to contribute a lot to the performance improvement, making it unclear that how much the teacher prior knowledge indeed contributes.
- The novelty is not significant. Using masked teacher features as target for training a student model has been explored by existing works, like BEIT-V3. Although the concrete setting is different, the idea is similar.

**Note From Pc:**

if the above contains the word "oral" or "spotlight" please see: "oral" presentation means -> notable-top-5% and "spotlight" means -> notable-top-25%. As stated in our emails, we are disassociating presentation type from AC recommendations

**Summary Of Ac-Reviewer Meeting:**

N/A